# Score-based Random Tomography of Dynamic Structures

## Abstract

A central problem in imaging sciences is the reconstruction of a three-dimensional object from two-dimensional projections. In many practical settings, however, the orientations at which projections are acquired cannot be controlled or observed. An additional challenge arises when the object is not static but undergoes continuous structural changes during data acquisition, a common characteristic of biomolecular systems imaged by cryo-electron microscopy. We propose a two-stage approach to random tomography of flexible objects that exhibit such structural variability. First, we learn a score-based diffusion model directly from 2D projection images, capturing the distribution of object conformations under unknown and randomly distributed orientations. Second, we reconstruct 3D volumes whose projections are consistent with the probability distribution implicitly defined by the learned diffusion model. Combined, these two stages enable 3D reconstruction in the presence of both unknown orientations and intrinsic structural dynamics.

## 1. Introduction

To reconstruct a 3D structure from a set of 2D projections is a central problem in imaging sciences. In standard tomography, projections correspond to line integrals of the object's density measured from multiple orientations. Recovering the underlying 3D structure from these indirect and often noisy observations is a classical inverse problem with applications in medical imaging, materials science, and biological microscopy (Kak & Slaney, 2001; Deans, 2007; Frank, 2006).

However, in many practical settings the orientations at which projections are acquired cannot be controlled or directly observed. This is the case, for example, in cryo-electron

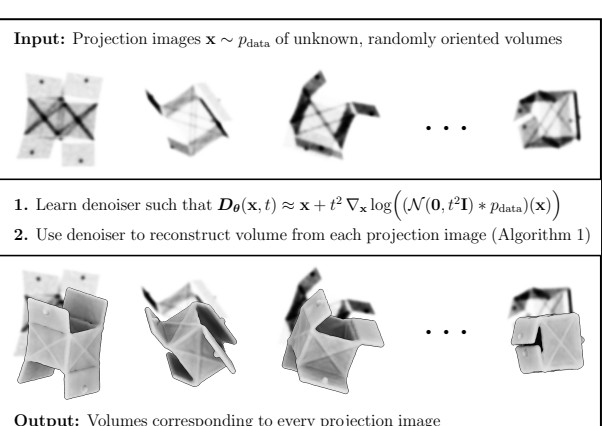

*Figure 1.* Our approach to random tomography in the presence of structural dynamics. Top row: example 2D projection images (here, projections of a dynamically moving box). Bottom row: the corresponding 3D volumes that generated each projection. To recover these volumes, we first learn a representation of the projections by training a denoiser $\mathbf{D}_{\boldsymbol{\theta}}(\mathbf{x}, t)$. This learned representation is then used to reconstruct the 3D structure corresponding to each input image.

microscopy (cryo-EM) (Frank, 2006; Nogales & Scheres, 2015), where individual macromolecular complexes are imaged in unknown and randomly distributed poses. The resulting uncertainty in viewing directions fundamentally complicates the reconstruction task, as both the 3D structure and the associated projection orientations must be inferred jointly from the observed data. In this setting, only the projection images are available for reconstruction; their relative orientations are unobserved, and no prior information about the underlying volume is assumed. This challenging inverse problem is commonly referred to as *random tomography* (Panaretos, 2009).

Standard formulations of random tomography assume that all projection images arise from a single, static 3D object. In many applications, however, the imaged object is inherently dynamic, introducing an additional layer of complexity. Here, we address the resulting challenge of heterogeneous random tomographic reconstruction. Given a collection of 2D projection images, the goal is to infer the underlying 3D densities that generated these observations. Crucially, we do not assume that all projections correspond to an identical volume; instead, the object may vary over time or composi-

---
[1]Anonymous Institution, Anonymous City, Anonymous Region, Anonymous Country. Correspondence to: Anonymous Author <anon.email@domain.com>.

**Input:** Projection images $\mathbf{x} \sim p_{\text{data}}$ of unknown, randomly oriented volumes

1. Learn denoiser such that $\boldsymbol{D}_{\boldsymbol{\theta}}(\mathbf{x}, t) \approx \mathbf{x} + t^2 \nabla_{\mathbf{x}} \log\Big( (\mathcal{N}(\mathbf{0}, t^2\mathbf{I}) * p_{\text{data}})(\mathbf{x}) \Big)$

2. Use denoiser to reconstruct volume from each projection image (Algorithm 1)

**Output:** Volumes corresponding to every projection image

tion, giving rise to multiple structures across different images. This setting arises naturally in cryo-EM, where large ensembles of biomolecules are rapidly frozen and imaged in unknown and randomly distributed orientations, producing noisy 2D projection images of individual particles. Owing to thermal motion and biochemical activity, these particles frequently occupy a heterogeneous set of conformations, resulting in substantial structural variability in the observed data.

Several deep learning-based methods have been developed to address heterogeneous random tomography in cryo-EM including, for example VAE-based methods such as Cryo-DRGN, CryoDRGN2 and cryoFIRE (Zhong et al., 2021a;b; Levy et al., 2022a). In this work, we introduce a fundamentally different approach to random tomography in the presence of structural dynamics that departs from existing methods based on explicit generative modeling of 3D volumes. Instead, our framework leverages score-based generative models (Song et al., 2020) to learn a neural representation of the 2D projection data. More precisely, we learn a denoiser $\mathbf{D}_{\boldsymbol{\theta}}(\mathbf{x}, t)$ to approximate the score of the mollified data distribution $p_{\text{data}}(\mathbf{x})$ defined over the 2D projections (Karras et al., 2022). This learned representation captures the distribution of projections under unknown orientations and structural variability, which we then exploit to reconstruct full 3D volumes from each projection (see Figure 1). By decoupling the modeling of 2D data from 3D reconstruction, our method provides a flexible and scalable alternative to existing approaches such as CryoDRGN and cryoFIRE, enabling reconstruction of highly heterogeneous structures from projection images.

## 2. Background

### 2.1. Score-Based Diffusion Models

The first step of our approach is to model the distribution of a set of samples $\{\mathbf{x}_{\text{data}}^i\}$ (in our case 2D projection images) drawn from an unknown distribution $p_0$, which we also denote as $p_{\text{data}}$. In score-based diffusion models (SBDMs), one constructs a family of noise-perturbed distributions (Song et al., 2020):

$$p_t(\mathbf{x}) = \int \mathcal{N}(\mathbf{x}; \mathbf{x}_0, t^2\mathbf{I})\, p_0(\mathbf{x}_0)\, \mathrm{d}\mathbf{x}_0 \,,$$

which can be interpreted as progressively "mollifying" the target distribution $p_0$ by convolving it with a Gaussian: $p_t = \mathcal{N}(\mathbf{0}, t^2\mathbf{I}) * p_0$. Here, we adopt the noise schedule proposed by Karras et al. (2022), meaning that we add Gaussian noise to the training data for $t > 0$. As $t$ increases towards some large $T$, the signal-to-noise ratio of the samples decreases, and $p_t$ gradually converges towards an isotropic Gaussian, $\mathbf{x} \sim \mathcal{N}(\mathbf{0}, T^2\mathbf{I})$.

In SBDMs, new samples are generated using the score of

the mollified distribution, $\nabla_{\mathbf{x}} \log p_t(\mathbf{x})$. Direct computation of this score is generally intractable. Instead, we learn a parameterized surrogate model $\nabla_{\mathbf{x}} \log p_{\boldsymbol{\theta},t}(\mathbf{x})$, which can be expressed in terms of a denoiser as $\nabla_{\mathbf{x}} \log p_{\boldsymbol{\theta},t}(\mathbf{x}) = (\mathbf{D}_{\boldsymbol{\theta}}(\mathbf{x}, t) - \mathbf{x})/t^2$. Learning the denoiser $\mathbf{D}_{\boldsymbol{\theta}}$ involves optimizing its parameters $\boldsymbol{\theta}$ with denoising score matching (Vincent, 2011; Song et al., 2020; Karras et al., 2022):

$$\min_{\boldsymbol{\theta}} \quad \mathbb{E}_{\substack{t \sim q \\ \mathbf{x} \sim p_0 \\ \boldsymbol{\epsilon} \sim \mathcal{N}(\mathbf{0}, \mathbf{I})}} \left\| \mathbf{x} - \mathbf{D}_{\boldsymbol{\theta}}(\mathbf{x} + t\boldsymbol{\epsilon}, t) \right\|^2 \tag{1}$$

where $q$ is the distribution of the noise level $t$. After training, we can use the learned denoiser to generate synthetic images $\mathbf{x}_{t_{\min}}$ (with a small, positive $t_{\min} \approx 0$) by integrating the velocity field starting from $\mathbf{x}_T \sim \mathcal{N}(\mathbf{0}, T^2\mathbf{I})$ at $t = T$ and solving the ODE $\mathrm{d}\mathbf{x}/\mathrm{d}t = -t\,\nabla_{\mathbf{x}} \log p_{\boldsymbol{\theta},t}(\mathbf{x})$ backward in time until $t = t_{\min}$. In our experiments, we set $t_{\min} = 0.002$, $T = 80$, and normalize the input/training data to zero mean and unit variance.

### 2.2. Score Distillation Sampling

The denoiser can be utilized to recover a 3D structure via score distillation sampling (SDS) (Poole et al., 2022), which enables us to learn 3D models whose rendered views resemble samples from the data distribution. Let $\phi$ denote the parameters of a 3D volume and $\mathbf{g}$ a differentiable volume renderer. SDS optimizes $\phi$ by defining a loss $\mathcal{L}_{\text{SDS}}$ on the rendered images $\mathbf{g}(\phi)$, whose gradient $\nabla_{\phi} \mathcal{L}_{\text{SDS}}(\boldsymbol{\theta}, \mathbf{g}(\phi))$ is given by

$$\mathbb{E}_{t,\boldsymbol{\epsilon}} \left[ w(t) \left( \boldsymbol{\epsilon}_{\boldsymbol{\theta}}(\mathbf{g}(\phi), y, t) - \boldsymbol{\epsilon} \right) \frac{\partial \mathbf{g}(\phi)}{\partial \phi} \right], \tag{2}$$

where $t \sim \mathcal{U}([a, b])$, $\boldsymbol{\epsilon} \sim \mathcal{N}(\mathbf{0}, \mathbf{I})$, and $w(t)$ weights the contribution of the gradient per timestep. In the original formulation of SDS, $\boldsymbol{\epsilon}_{\boldsymbol{\theta}}(\mathbf{x}, y, t) = (\mathbf{x} - \mathbf{D}_{\boldsymbol{\theta}}(\mathbf{x}, y, t))/t$ predicts the noise component present in the input $\mathbf{x}$, conditioned on text embeddings $y$.

SDS has been applied to generate 3D models based on neural radiance fields (NeRF) (Poole et al., 2022) or Gaussian splats (Tang et al., 2023). These approaches used text-to-image diffusion models trained on internet-scale datasets of natural images paired with text descriptions. A key advantage of this approach is its ability to exploit the abundance of large-scale 2D image data, in contrast to the relative scarcity of annotated 3D training data.

Although the 3D assets generated by Poole et al. (2022) and Tang et al. (2023) are not volumetric densities but renderable surfaces learned from large-scale datasets of natural images paired with text descriptions, the underlying problem formulation extends to cryo-EM. In cryo-EM, large amounts of 2D projection data are readily available, while

the corresponding 3D structural information is entirely unobserved in the *ab initio* setting. Crucially, the cryo-EM forward model, the projection of a 3D volume to 2D images, is differentiable, enabling us to use score distillation to optimize volumetric densities such that their projections match the learned distribution of experimental images.

### 2.3. Cryo-EM

In real space, the formation of a cryo-EM image can be modeled as follows. Every raw image $\mathbf{p}_n$ corresponds to a 2D projection $\Pi$ of a 3D volume $\mathbf{V}_n$ (also called a *particle*) that has been rotated by $\mathcal{R}_n$. The projection is subsequently shifted by $\mathcal{T}_n$, convolved with a point spread function $\text{PSF}_n$ and contaminated by an additive noise term $\boldsymbol{\epsilon}_n$ (typically Gaussian):

$$\mathbf{p}_n = \text{PSF}_n * \big(\mathcal{T}_n \circ \Pi \circ \mathcal{R}_n\big)(\mathbf{V}_n) + \boldsymbol{\epsilon}_n. \tag{3}$$

In the heterogeneous formulation of the forward model (Eq. 3), each raw particle image $\mathbf{p}_n$ is characterized not only by its own point spread function, in-plane translation, rotation, and noise realization, but also by an individual underlying 3D volume $\mathbf{V}_n$, where $n$ is a particle-specific index. In the considerably simpler homogeneous setting, all particle images are assumed to arise from a single, identical conformation, such that $\mathbf{V}_n \equiv \mathbf{V}$ for all $n$.

Every particle image $\mathbf{p}_n$ is cropped from a micrograph containing many other particles and typically exhibits a low signal-to-noise ratio (SNR). To enhance SNR, cryo-EM software such as RELION (Scheres, 2012), cryoSPARC (Punjani et al., 2017), EMAN2 (Tang et al., 2007), SIMPLE3 (Caesar et al., 2020), and ASPIRE (Zhao & Singer, 2014) incorporate procedures to estimate the PSF or rather its Fourier transform, the CTF (contrast transfer function), correct for CTF effects, and combine groups of similar raw particle images into 2D class-averages. In RELION and cryoSPARC, these 2D class-averages are used primarily as a screening tool to identify and discard particle images corresponding to junk or contamination. In contrast, SIMPLE, EMAN, and ASPIRE additionally use class-averages directly for 3D reconstruction. However, class-average-based *ab initio* reconstruction implemented in SIMPLE, EMAN, and ASPIRE assumes that the class-averaged data are homogeneous, and therefore these methods recover only a single volume for each input dataset.

In recent years, deep learning–based methods for heterogeneous refinement, such as CryoDRGN (Zhong et al., 2019), 3DFlex, and OPUS-DSD (Luo et al., 2023; 2024), as well as fully *ab initio* reconstruction approaches such as CryoDRGN2 (Zhong et al., 2021a) and CryoFIRE (Levy et al., 2022b), have attracted increasing interest. These frameworks aim to perform heterogeneous 3D reconstruction directly from stacks of raw particle images, using associated CTF parameters together with initial estimates of particle orientations and in-plane shifts derived from a preceding homogeneous consensus refinement. Notably, CryoDRGN2 and CryoFIRE also include *ab initio* modes, allowing reconstruction without any precomputed rotational or translational alignment information.

### 3. Methods

In the cryo-EM setting, our approach operates on 2D class-averages, which exhibit a high SNR and are largely free of PSF–induced blurring. However, these class-averages are available in far smaller numbers than the $10^4$–$10^7$ raw particle images typically acquired in cryo-EM experiments. Consequently, for each system we are limited to a relatively small set of 2D projections, from which we seek to infer the underlying 3D volumes that gave rise to the observed data.

We first train a diffusion model on 2D projection images. Prior to training, all images are normalized to have zero mean and unit variance. In addition, we apply data augmentation consisting of random affine transformations, including in-plane rotations, flips, and translations of up to 1% of the image size. To mitigate rotation-induced artifacts, we use bilinear interpolation and apply circular masking to all images. Training follows the schedule proposed by Karras et al. (2022).

After learning the 2D diffusion model, our goal is to recover the 3D volumes $\mathbf{V}$ corresponding to each input image. Each volume is represented as a 3D voxel grid $\mathbf{V}$. For a given rotation $\mathbf{R}$, volume $\mathbf{V}$ induces a mollified distribution over 2D projection images

$$q_t(\cdot \,|\, \mathbf{V}, \mathbf{R}) := \mathcal{N}(\mathbf{f}(\mathbf{V}, \mathbf{R}), t^2\mathbf{I}).$$

To obtain a sample from $q_t(\cdot \,|\, \mathbf{V}, \mathbf{R})$, we use the forward operator $\mathbf{f}(\mathbf{V}, \mathbf{R})$, which first rotates the volume and projects it onto the 2D plane, after which we add a Gaussian noise term with a covariance matrix $t^2\mathbf{I}$. To perform 3D reconstruction, we seek a volume $\mathbf{V}$ that minimizes the reverse KL divergence, averaged over all possible rotations, between the mollified distribution of volume projections $q_t(\cdot \,|\, \mathbf{V}, \mathbf{R})$ and the parametric model of the mollified data distribution $p_{\boldsymbol{\theta},t}$:

$$\min_{\mathbf{V}} \int_{\text{SO}(3)} \mathrm{D}_{\text{KL}}\left(q_t(\cdot \,|\, \mathbf{V}, \mathbf{R}) \,\|\, p_{\boldsymbol{\theta},t}\right) \mathrm{d}\mathbf{R} \tag{4}$$

Here, $t > 0$ controls the amount of mollification. We use stochastic gradient descent (SGD) to determine the optimal volume $\mathbf{V}$. The gradient of the objective in Eq. (4) is, up to a constant factor, given by

$$\mathop{\mathbb{E}}_{\substack{\mathbf{R}\sim\text{SO}(3) \\ \boldsymbol{\epsilon}\sim\mathcal{N}(\mathbf{0},\mathbf{I})}} \left[\left(\frac{\partial \mathbf{f}(\mathbf{V}, \mathbf{R})}{\partial \mathbf{V}}\right)^{\!\top}\!\big(\mathbf{f}(\mathbf{V}, \mathbf{R}) - \mathbf{D}_{\boldsymbol{\theta}}(\mathbf{f}(\mathbf{V}, \mathbf{R}) + t\boldsymbol{\epsilon}, t)\big)\right].$$

$$\tag{5}$$

A detailed proof showing that this gradient corresponds to the objective in Eq. (4) is provided in the Appendix. The derivation of the objective is inspired by Eq. (4) in Poole et al. (2022). During optimization, the parameters $\boldsymbol{\theta}$ of the denoiser are kept fixed. Algorithm 1 summarizes the reconstruction procedure, which applies SGD to optimize the voxel values of the volume. We deliberately employ a simple SGD scheme without an adaptive learning rate. Moreover, we avoid automatic differentiation and instead implement both the forward and backward passes manually (see Code Listings 1, 2, and 3 in the Appendix).

Although our method is inspired by SDS (Poole et al., 2022), it differs fundamentally from previous applications. Rather than relying on internet-scale text-image pairs, our diffusion model is trained on projection data from a single experiment imaging a specific structure, such as a macromolecular complex, using on the order of 30 to 300 images. Moreover, our objective is not to generate visually plausible surface renderings for 3D animation, but to reconstruct a physically meaningful 3D density map of the imaged biomolecular complex that also captures subsurface information. Accordingly, we represent 3D structures using voxel grids, the standard volumetric representation in cryo-EM, rather than NeRFs or Gaussian splats.

### 3.1. Heterogeneous Reconstruction by Mode Seeking

Heterogeneous class-average datasets contain projection images of a macromolecular complex in multiple conformational states. As a result, any single 3D reconstruction can explain only a subset of these class-averages. When reconstructing a particular conformation, the goal is therefore to capture only a subset of the modes present in the overall distribution of class-averages. Our method is naturally aligned with this objective: the proposed loss encourages a volume to generate projections that attain a high probability under the data distribution, without requiring it to simultaneously match all of the class-averages seen during training.

This distinction is made explicit by the optimization problem in Eq. (4). The reverse KL divergence emphasizes fitting selected modes of the modeled distribution $p_{\boldsymbol{\theta},t}$, rather than covering it entirely. We view this property as a key advantage over maximum-likelihood-based approaches commonly used in cryo-EM reconstruction. These approaches minimize the forward KL divergence and therefore force a single volume to explain all class-averages, a requirement that is fundamentally incompatible with heterogeneous data.

To ensure that all conformations present in the data can, in principle, be recovered, we reconstruct a separate 3D volume for each class-average used for training the denoiser. To encourage that the $i$-th volume captures the conformation depicted by the corresponding class-average $\mathbf{x}_{\text{data}}^i$, we impose a soft constraint that aligns the projection of the volume

with the observed image such that ideally $\mathbf{x}_{\text{data}}^i = \mathbf{f}(\mathbf{V}, \mathbf{I})$ where $\mathbf{I}$ denotes the identity matrix. A detailed description of the resulting optimization procedure is provided in Algorithm 1. For a specific illustration of how Algorithm 1 iteratively reconstructs a single 3D conformation from the capsid-model dataset (described in the next section), refer to Figure 8 in the Appendix.

---

**Algorithm 1** 3D reconstruction

**Input:** denoiser $\mathbf{D}_{\boldsymbol{\theta}}$, base image $\mathbf{x}_{\text{data}}^i$ (size $\times$ size pixels), number of steps $S$, step-dependent batch sizes $M_1, \ldots, M_S \geq 2$, timestep $t \geq t_{\min}$, learning rate $\alpha > 0$
*// initialize volume*
$\mathbf{V}[x,y,z] \leftarrow 0 \; \forall x,y,z \in \{1,\ldots,\text{size}\}$
**for** $s \in \{1,\ldots,S-1\}$ **do**
  *// forward projection*
  $\mathbf{R}_m \sim \mathcal{U}(\text{SO}(3)) \; \forall m \in \{2,\ldots,M_s\}$
  $\mathbf{R}_1 \leftarrow \mathbf{I}$
  $\mathbf{x}_m \leftarrow \mathbf{f}(\mathbf{V}, \mathbf{R}_m) \; \forall m \in \{1,\ldots,M_s\}$
  *// compute residuals*
  $\boldsymbol{\epsilon}_m \sim \mathcal{N}(\mathbf{0}, \mathbf{I}) \; \forall m \in \{2,\ldots,M_s\}$
  $\mathbf{r}_m \leftarrow \mathbf{x}_m - \mathbf{D}_{\boldsymbol{\theta}}(\mathbf{x}_m + t\boldsymbol{\epsilon}_m, t) \; \forall m \in \{2,\ldots,M_s\}$
  $\mathbf{r}_1 \leftarrow \mathbf{x}_1 - \mathbf{x}_{\text{data}}^i,$
  *// back project residuals*
  $\Delta\mathbf{V} \leftarrow \texttt{backproject}(\mathbf{r}_m, \mathbf{R}_m)$
  *// update volume*
  $\mathbf{V} \leftarrow \mathbf{V} - \alpha\Delta\mathbf{V}$
**end for**
**Return:** volume $\mathbf{V}$ corresponding to $\mathbf{x}_{\text{data}}^i$

---

## 4. Experiments

We conducted experiments on both real and synthetic class-average datasets spanning a wide range of characteristics (see Figure 2). In particular, we considered datasets varying in the number of class-averages, image size, symmetry properties of the underlying 3D structures, and the degree of structural homogeneity versus heterogeneity. Across all experiments, with the exception of the ribosome dataset, we used a fixed denoiser architecture.

For the denoiser architecture, we employed the `UNet2DModel` (Ronneberger et al., 2015) from the Diffusers library (von Platen et al., 2022). The network consists of three downsampling blocks (`DownBlock2D`, `DownBlock2D`, `AttnDownBlock2D`) with channel sizes 32, 64, and 128, respectively, followed by three corresponding upsampling blocks (`AttnUpBlock2D`, `UpBlock2D`, `UpBlock2D`). The bottleneck of the UNet was kept at its default configuration. Overall, the denoiser comprises approximately 3.9 million parameters.

For the ribosome dataset, we increased the network capacity

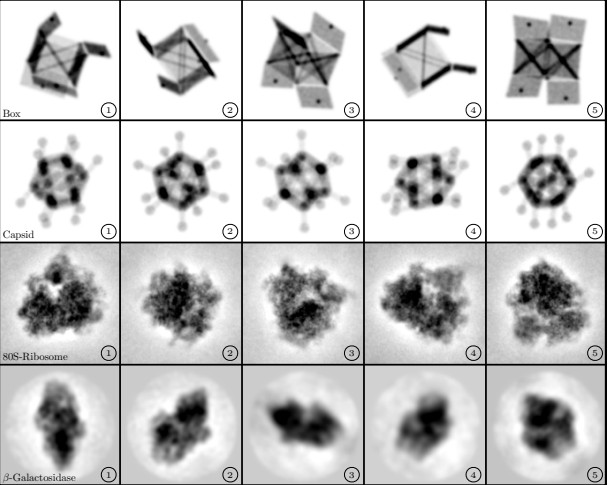

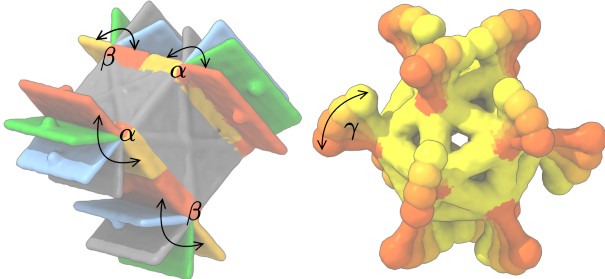

*Figure 3.* **Left:** The conformation of the box model is parameterized by two continuous angles that rotate the lids. The two upper lids are rotated as well as their mirrored counterparts at the bottom of the box. **Right:** The capsid model exhibits a continuous transition of the outer "arms" from an open (red) to a closed state (yellow), controlled by a single parameter.

*Figure 2.* First five class-averages of each test dataset. The five images of the box-model (shown in the top row) serve as the basis for the reconstructions displayed on the left-hand side of Fig. 4. The 3D volumes shown on the left-hand side of Fig. 5 are recovered from the five images of the capsid-model shown in the second row. For the homogeneous datasets showing the 80S ribosome (row three) and $\beta$-galactosidase (row four), we generated only a single reconstruction, derived from the first image of each dataset.

by adding an additional block to both the encoder and decoder, and adjusted the channel sizes to 32, 64, 96, and 128, resulting in approximately 5.3 million parameters. Training of the denoiser followed the procedure described by Karras et al. (2022). Additional training details and hyperparameters are provided in Table 2 in the Appendix.

We benchmarked our method against state-of-the-art class-average–based reconstruction approaches in cryo-EM, specifically those implemented in the software packages SIMPLE3 (Caesar et al., 2020) and EMAN2 (Tang et al., 2007). For the homogeneous ribosome dataset, which contains a large number of particle images, we additionally evaluated a standard common-lines reconstruction algorithm as implemented in ASPIRE (Shkolnisky & Singer, 2012; Singer et al., 2010; Andén & Singer, 2018). To facilitate comparison in the heterogeneous setting, we adapted Cryo-DRGN2 (Zhong et al., 2021b) to operate on class-averages by replacing their CTFs with an identity operator (see Appendix B.2 for more information).

For evaluation, we use the Fourier Shell Correlation (FSC) between the reconstructed 3D volume and the corresponding ground truth, which is the standard metric for assessing reconstruction quality in cryo-EM. Prior to computing FSC, each reconstruction is aligned to its ground-truth volume. Alignment is performed using UCSF ChimeraX (Meng et al., 2023): we employ the `fitmap` tool to determine the optimal orientation and the `resample` function to interpolate the reconstructed volume onto the voxel grid of the ground

truth. Since reconstruction from projection images is inherently ambiguous up to a global mirroring, we account for this ambiguity during alignment.

### 4.1. Heterogeneous reconstruction

To test how well our method performs on datasets exhibiting strong structural heterogeneity, we designed two flexible 3D objects (Figure 3). The first object is a box equipped with four lids that have movable joints. The second structure is a virus-like capsid with flexible spikes/arms on its exterior. Table 1 provides further details on the test datasets.

To generate a 2D dataset from a movable 3D object, we sampled for each synthetic class-average a random conformation, which is randomly rotated and projected along the first axis onto the 2D plane. We sampled 300 projection images of the box and 200 of the capsid-model. Each projection image has a shape of $88 \times 88$ pixels. By design, each movable 3D object admits a continuous range of conformations. Consequently, every class-average corresponds to a distinct conformation of the same underlying 3D object.

For each dataset, we generated reconstructions for every synthetic class-average using Algorithm 1, resulting in 300 box volumes and 200 capsid volumes. The settings and algorithmic parameters are detailed in Table 2 in the Appendix. In contrast to our approach, CryoDRGN2 learns a continuous latent space of possible conformations. Its variational auto-encoder (Kingma & Welling, 2013) uses an encoder to embed the images into this latent conformation space, and a decoder that maps latent coordinates back to 3D volumes. In this way, CryoDRGN2 establishes a correspondence between each image and the underlying conformation it depicts. Accordingly, we also obtained 300 box volumes and 200 capsid volumes with CryoDRGN2.

To assess the quality of all reconstructions, we computed FSC against the ground-truth volumes. Such validation is only feasible for synthetic datasets, where both the true

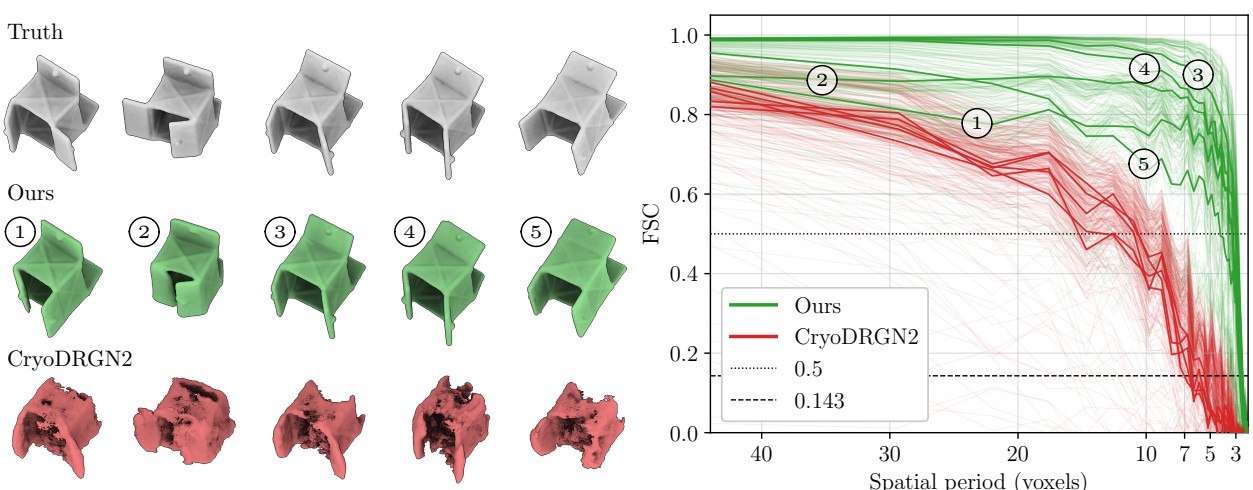

*Figure 4.* **Left:** 3D reconstructions of the box model corresponding to the first five projection images in the dataset generated by our method (green) and CryoDRGN2 (red). The first row displays the ground truth volumes that were used to generate these five projections (see the first row of Figure 2). **Right:** FSC curves comparing the 300 ground truth volumes with the reconstructions obtained with our method and CryoDRGN2, showing that our approach can reconstruct each volume at a higher resolution than CryoDRGN2. Note that reconstructions 1 and 2 illustrate examples where our method fails: although the reconstructed boxes appear plausible, the lid positions are partially incorrect.

3D model and the exact image-to-conformation mapping are known. The results obtained on the two heterogeneous datasets are shown in Figures 4 and 5. Both heterogeneous datasets are highly challenging as they offer only sparse information about the underlying volumes, with just a single image available for each conformation. In both cases, we observe that our method can consistently reconstruct a wide range of plausible 3D conformations that exhibit strong correlation with the ground truth across many spatial frequencies. In contrast, CryoDRGN2 has difficulty producing *ab initio* reconstructions for both datasets, and this issue is especially pronounced for the capsid dataset.

It is important to emphasize that CryoDRGN2 is intended for *ab initio* reconstruction from very large collections of noisy images, rather than from a small number of high-SNR images. However, this limitation does not appear to affect performance in the homogeneous test scenarios (see next section), where CryoDRGN2 performs comparably to other reconstruction algorithms. We hypothesize that CryoDRGN2 has difficulties facing the joint challenge of estimating accurate image-to-volume orientations while simultaneously handling structural heterogeneity. In particular, the capsid's core is highly symmetric, producing a large set of candidate orientations with similarly low loss, since projections along many different directions can match the capsid volume equally well (i.e., the core appears nearly identical from multiple viewing angles). This multi-modal

optimization landscape makes it difficult to identify the correct orientations during CryoDRGN2's *pose search*. The box model shows a similar behavior, but has fewer symmetries. In this case, CryoDRGN2's performance is closer to that of our method. Our approach views the volume from a randomly chosen orientation and then asks the denoiser to produce a more plausibly "box"-like or "capsid"-like version of the resulting projection. As a result, our method does not involve any pose search matching a volume to a class-average, nor does it encounter the associated challenges.

Let us highlight some pitfalls of our approach. The first two box reconstructions shown in Fig. 4 appear to be plausible. The inferred motion, however, is incorrect, because the lower part of the box should mirror the motion of the upper part. Such errors are difficult to detect in the absence of a ground truth and may result in incorrect dynamical interpretations of the specimen under study. Another type of error is easier to detect, but less frequent: in these cases, the core is largely reconstructed correctly, but the movable parts are only partially recovered (Supplementary Figure 9).

We additionally employed SIMPLE, ASPIRE, and EMAN to reconstruct a consensus structure for the box and capsid datasets. SIMPLE and ASPIRE did not recover any meaningful 3D structure that exhibited similarity to the ground truth. EMAN, on the other hand, succeeded in approximating the central core of both models but did not recover the flexible parts (Supplementary Figure 10).


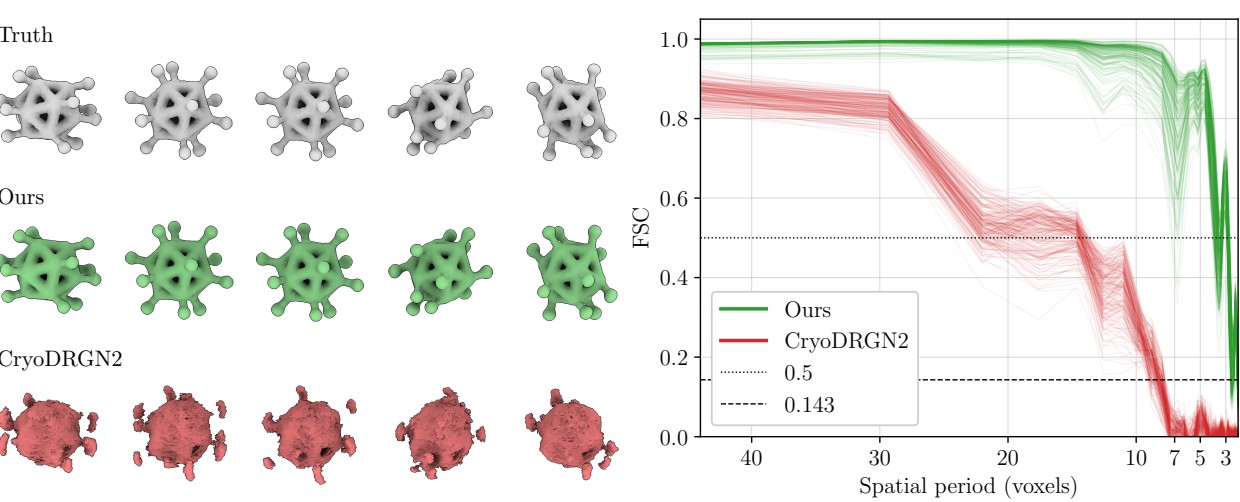

*Figure 5.* **Left:** 3D reconstructions of the capsid model associated with the first five projection images generated with our method (green) and CryoDRGN2 (red). We highlighted the respective FSC curves for the five examples on the right. The top row shows the ground truth volumes, i.e. random conformations of the capsid model, used to generate these first five projections in the dataset (see the second row of Figure 2). **Right:** FSC curves between the 200 ground truth volumes and the reconstructions produced by our method and CryoDRGN2. We observe that CryoDRGN2 has substantial difficulty reconstructing even low-frequency features, whereas our method is able to recover each volume at high resolution.

## 4.2. Homogeneous reconstruction

We evaluated our method using two homogeneous datasets that were acquired experimentally. The first dataset comprises 100 class-averages generated by SIMPLE using EMPIAR-10028 micrographs of the 80S ribosome, as provided in the SIMPLE tutorial. These images have dimensions of $136 \times 136$ pixels and a sampling rate of 2.68 Å/pixel. We evaluate our *ab initio* reconstructions, as well as those from competing approaches, by comparing them against the refined volume from EMD-2660. The second dataset, taken from the RELION tutorial (RELION Development Team), consists of 34 class-averages of $\beta$-galactosidase (generated from EMPIAR-10204 micrographs) with a sampling rate of 3.54 Å/pixel and size $64 \times 64$. The refined volume from EMD-6840 serves as ground truth. Supplementary Table 1 provides an overview of these test datasets.

We ran Algorithm 1 once for each molecular complex selecting the first image in each dataset as the base image (see Supplementary Table 2 for details). The results are shown in Figures 6 and 7. Compared to the other reconstruction methods, our approach achieves superior performance on the $\beta$-galactosidase dataset, but underperforms on the 80S ribosome dataset. We attribute the improved performance on $\beta$-galactosidase primarily to the D2 symmetry of the tetramer. The competing methods assume that each class-average corresponds to a unique pose, whereas our method does not perform an explicit pose search. Informally, in Al-

gorithm 1, the denoiser "views" the volume from a random orientation and proposes an updated volume that makes the projection it sees resemble more closely an image from the training data distribution. Consequently, the denoiser can exploit information learned from a single class-average to refine the volume from all orientations that yield similar projections. This mechanism, however, is effective only for volumes exhibiting symmetry. In the case of the fully asymmetric 80S ribosome, this advantage disappears, and classical approaches that explicitly search over poses for each class-average appear to be superior.

## 5. Conclusion and Outlook

In this work, we have shown that score models trained solely on 2D projection images can be used for 3D reconstruction without any prior knowledge of viewing orientations, and can even handle volumes whose shapes change dynamically from image to image. We demonstrated that the proposed approach achieves remarkable performance in reconstructing highly dynamic structures from small input dataset of high-SNR images. This central result lays the groundwork for further developments of the method. Going forward, we plan to extend our approach to enable *ab initio* 3D reconstruction of heterogeneous volumes from large collections of noisy, PSF-corrupted projections. We anticipate that this will open up a fundamentally new direction for tackling the heterogeneous reconstruction problem in cryo-EM.

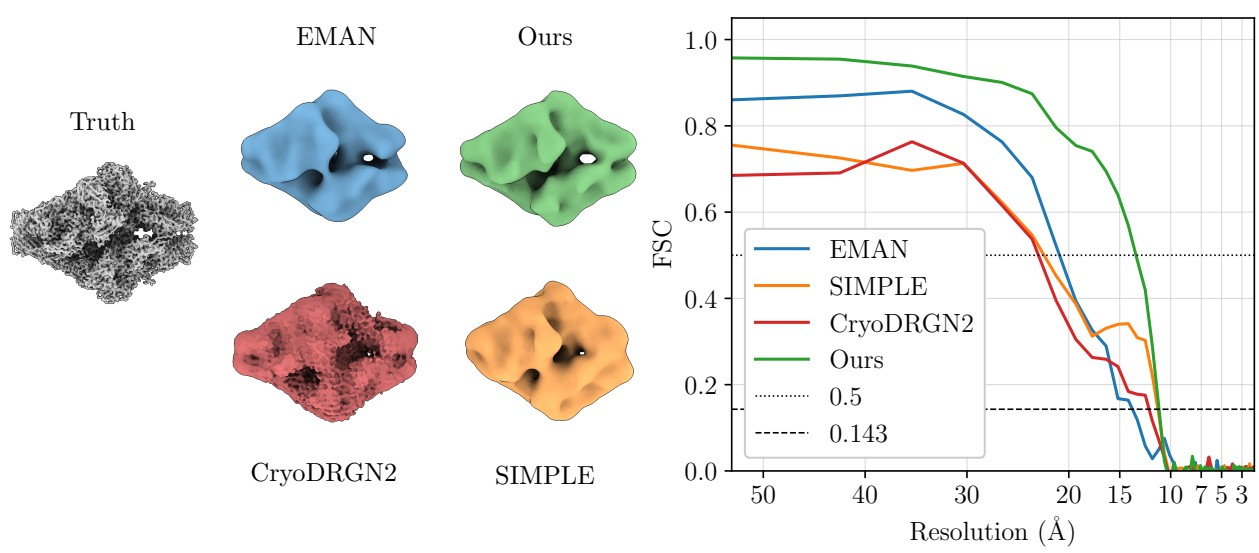

*Figure 6.* **Left:** 3D reconstructions of $\beta$-galactosidase produced by our method and by several standard approaches in cryo-EM, together with the ground truth (EMD-6840). **Right:** FSC curves comparing reconstructed volumes with the ground truth clearly demonstrate that our method equals or exceeds the others approaches in correlation across the a wide range of resolutions.

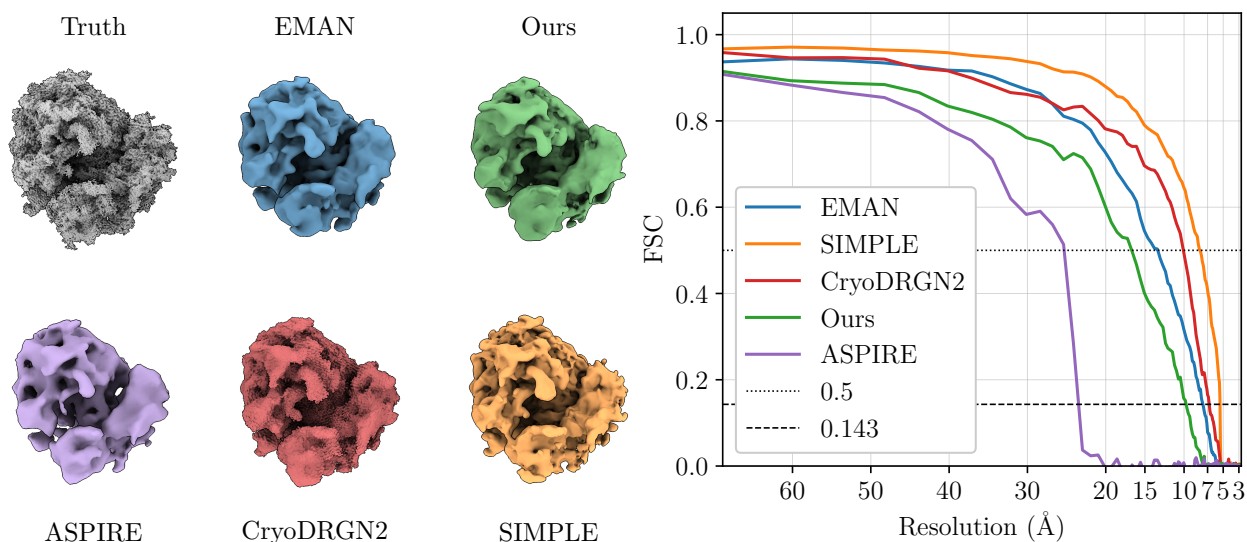

*Figure 7.* **Left:** 3D reconstructions of the 80S ribosome generated by the various reconstruction methods alongside the ground truth density (EMD-2660). All methods yield a reasonable *ab initio* reconstruction. **Right:** FSC curves revealing that our method performs slightly worse than the other methods, with the exception of ASPIRE's common lines approach.

## Impact Statement

This paper presents work whose goal is to advance the field of Machine Learning. There are many potential societal consequences of our work, none which we feel must be specifically highlighted here.

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
