## A. Methods

### A.1. Derivation of the Gradient

**Proof that the gradient of (4) is proportional to (5)** Per construction, we can write $\mathbf{D}_{\boldsymbol{\theta}}(\mathbf{x}, t) = \mathbf{x} + t^2 \nabla_{\mathbf{x}} \log p_{\boldsymbol{\theta},t}(\mathbf{x})$ (Karras et al., 2022). So the gradient can be written as

$$\underset{\substack{\mathbf{R} \sim \text{SO}(3) \\ \boldsymbol{\epsilon} \sim \mathcal{N}(\mathbf{0},\mathbf{I})}}{\mathbb{E}} \left[ \left( \frac{\partial \mathbf{f}(\mathbf{V},\mathbf{R})}{\partial \mathbf{V}} \right)^{\top} \left( \mathbf{f}(\mathbf{V},\mathbf{R}) - \mathbf{D}_{\boldsymbol{\theta}}(\mathbf{f}(\mathbf{V},\mathbf{R}) + t\boldsymbol{\epsilon}, t) \right) \right] \tag{6}$$

$$= - \underset{\substack{\mathbf{R} \sim \text{SO}(3) \\ \boldsymbol{\epsilon} \sim \mathcal{N}(\mathbf{0},\mathbf{I})}}{\mathbb{E}} \left[ \left( \frac{\partial \mathbf{f}(\mathbf{V},\mathbf{R})}{\partial \mathbf{V}} \right)^{\top} \left( t^2 \nabla \log p_{\boldsymbol{\theta},t}(\mathbf{f}(\mathbf{V},\mathbf{R}) + t\boldsymbol{\epsilon}) + t\boldsymbol{\epsilon} \right) \right] \tag{7}$$

$$= - \nabla_{\mathbf{V}} \underset{\substack{\mathbf{R} \sim \text{SO}(3) \\ \boldsymbol{\epsilon} \sim \mathcal{N}(\mathbf{0},\mathbf{I})}}{\mathbb{E}} \left[ t^2 \log p_{\boldsymbol{\theta},t}(\mathbf{f}(\mathbf{V},\mathbf{R}) + t\boldsymbol{\epsilon}) + t\boldsymbol{\epsilon}^{\top} \mathbf{f}(\mathbf{V},\mathbf{R}) \right] \tag{8}$$

$$= - \nabla_{\mathbf{V}} \left( \underset{\substack{\mathbf{R} \sim \text{SO}(3) \\ \boldsymbol{\epsilon} \sim \mathcal{N}(\mathbf{0},\mathbf{I})}}{\mathbb{E}} \left[ t^2 \log p_{\boldsymbol{\theta},t}(\mathbf{f}(\mathbf{V},\mathbf{R}) + t\boldsymbol{\epsilon}) \right] + t \underbrace{\underset{\substack{\mathbf{R} \sim \text{SO}(3) \\ \boldsymbol{\epsilon} \sim \mathcal{N}(\mathbf{0},\mathbf{I})}}{\mathbb{E}} \left[ \boldsymbol{\epsilon}^{\top} \mathbf{f}(\mathbf{V},\mathbf{R}) \right]}_{0} \right) \tag{9}$$

$$= - t^2 \nabla_{\mathbf{V}} \underset{\substack{\mathbf{R} \sim \text{SO}(3) \\ \boldsymbol{\epsilon} \sim \mathcal{N}(\mathbf{0},\mathbf{I})}}{\mathbb{E}} \left[ \log p_{\boldsymbol{\theta},t}(\mathbf{f}(\mathbf{V},\mathbf{R}) + t\boldsymbol{\epsilon}) \right] \tag{10}$$

$$= - t^2 \nabla_{\mathbf{V}} \iint_{\text{SO}(3) \times \mathbb{R}^d} \mathcal{N}(\boldsymbol{\epsilon}; \mathbf{0}, \mathbf{I}) \log p_{\boldsymbol{\theta},t}(\mathbf{f}(\mathbf{V},\mathbf{R}) + t\boldsymbol{\epsilon}) \, \mathrm{d}\boldsymbol{\epsilon} \, \mathrm{d}\mathbf{R} \tag{11}$$

$$= - t^2 \nabla_{\mathbf{V}} \iint_{\text{SO}(3) \times \mathbb{R}^d} \mathcal{N}(\mathbf{x}; \mathbf{f}(\mathbf{V},\mathbf{R}), t^2\mathbf{I}) \log p_{\boldsymbol{\theta},t}(\mathbf{x}) \, \mathrm{d}\mathbf{x} \, \mathrm{d}\mathbf{R} \tag{12}$$

$$= t^2 \nabla_{\mathbf{V}} \left( \overbrace{\iint_{\text{SO}(3) \times \mathbb{R}^d} \mathcal{N}(\mathbf{x}; \mathbf{f}(\mathbf{V},\mathbf{R}), t^2\mathbf{I}) \log \mathcal{N}(\mathbf{x}; \mathbf{f}(\mathbf{V},\mathbf{R}), t^2\mathbf{I}) \, \mathrm{d}\mathbf{x} \, \mathrm{d}\mathbf{R}}^{\text{const. in } \mathbf{V}} - \iint_{\text{SO}(3) \times \mathbb{R}^d} \mathcal{N}(\mathbf{x}; \mathbf{f}(\mathbf{V},\mathbf{R}), t^2\mathbf{I}) \log p_{\boldsymbol{\theta},t}(\mathbf{x}) \, \mathrm{d}\mathbf{x} \, \mathrm{d}\mathbf{R} \right) \tag{13}$$

$$= t^2 \nabla_{\mathbf{V}} \int_{\text{SO}(3)} \mathrm{D}_{\text{KL}} \left( \mathcal{N}(\mathbf{f}(\mathbf{V},\mathbf{R}), t^2) \, \| \, p_{\boldsymbol{\theta},t} \right) \mathrm{d}\mathbf{R} \tag{14}$$

In this derivation, we used the following transformations:

1. Eq. (6) → Eq. (7): We plug in the definition of the denoiser: $\mathbf{D}_{\boldsymbol{\theta}}(\mathbf{x}, t) = \mathbf{x} + t^2 \nabla_{\mathbf{x}} \log p_{\boldsymbol{\theta},t}(\mathbf{x})$.

2. Eq. (7) → Eq. (8): We swap the gradient and the expectation, which are both linear operations. Moreover, we use the identities

$$\nabla_{\mathbf{V}} \log p_{\boldsymbol{\theta},t}(\mathbf{f}(\mathbf{V},\mathbf{R}) + t\boldsymbol{\epsilon}) = \left( \frac{\partial \mathbf{f}(\mathbf{V},\mathbf{R})}{\partial \mathbf{V}} \right)^{\top} \nabla \log p_{\boldsymbol{\theta},t}(\mathbf{f}(\mathbf{V},\mathbf{R}) + t\boldsymbol{\epsilon}), \quad \nabla_{\mathbf{V}} \boldsymbol{\epsilon}^{\top} \mathbf{f}(\mathbf{V},\mathbf{R}) = \left( \frac{\partial \mathbf{f}(\mathbf{V},\mathbf{R})}{\partial \mathbf{V}} \right)^{\top} \boldsymbol{\epsilon}$$

where $\partial \mathbf{f} / \partial \mathbf{V}$ is the partial Jacobian matrix of the forward operator (Jacobian with respect to $\mathbf{V}$).

3. Eq. (8) → Eq. (9): Linearity of expectation.

4. Eq. (9) → Eq. (10): Linearity of expectation.

5. Eq. (10) → Eq. (11): Definition of the expectation.

6. Eq. (11) → Eq. (12): Substitution $\boldsymbol{\epsilon} \to \mathbf{x} = \mathbf{f}(\mathbf{V},\mathbf{R}) + t\boldsymbol{\epsilon}$ such that $\boldsymbol{\epsilon} = (\mathbf{x} - \mathbf{f}(\mathbf{V},\mathbf{R}))/t$, $\mathrm{d}\boldsymbol{\epsilon} \to t^{-d}\mathrm{d}\mathbf{x}$ and

$$\mathcal{N}(\boldsymbol{\epsilon}; \mathbf{0}, \mathbf{I}) = \frac{1}{(2\pi)^{d/2}} e^{-\|\boldsymbol{\epsilon}\|^2/2} \to \frac{1}{(2\pi)^{d/2}} e^{-\|\mathbf{x} - \mathbf{f}(\mathbf{V},\mathbf{R})\|^2/2t^2} = t^d \mathcal{N}(\mathbf{x}; \mathbf{f}(\mathbf{V},\mathbf{R}), t^2\mathbf{I})$$

where $d$ is the dimension of image space (number of pixels).

7. Eq. (12) → Eq. (13): Introduction of a constant in $\mathbf{V}$ (negative information entropy of a Gaussian averaged over uniformly distributed rotations):

$$\iint_{\mathrm{SO}(3)\times\mathbb{R}^d} \mathcal{N}(\mathbf{x};\mathbf{f}(\mathbf{V},\mathbf{R}),t^2\mathbf{I})\log\mathcal{N}(\mathbf{x};\mathbf{f}(\mathbf{V},\mathbf{R}),t^2\mathbf{I})\,\mathrm{d}\mathbf{x}\,\mathrm{d}\mathbf{R} = -\frac{d}{2}(\log(2\pi t^2)+1)$$

8. Eq. (13) → Eq. (14): Definition of the KL divergence.

## A.2. Implementation Details

*Code 1*. Python implementation of the forward model.

```python
import torch
import torch.nn.functional as F

def forward_project(V, R):
    """
    Input:
        - V: shape (N, N, N)
        - R: shape (M, 3, 3)
    Returns:
        - projections: shape (M, N, N)
    """
    M = R.shape[0]
    N = volume.shape[-1]
    V_exp = V.reshape(1, 1, N, N, N).expand(M, -1, -1, -1, -1)
    orientation = torch.zeros((M,3,4), device=V.device)
    orientation[:, :3, :3] = R
    grid = F.affine_grid(orientation, V_exp.size(), align_corners=True)
    V_exp_rotated = F.grid_sample(V_exp,
                                  grid,
                                  align_corners=True)
    projections = V_exp_rotated.sum(dim=-3).squeeze(1)
    return projections
```

*Code 2*. Python implementation of the back projection.

```python
import torch
import torch.nn.functional as F

def back_project(residual, R):
    """
    Input:
        - residual: shape (M, N, N)
        - R: shape (M, 3, 3)
    Returns:
        - V: shape (N, N, N)
    """
    M = residual.shape[0]
    N = residual.shape[-1]
    V_exp_rotated = residual.view(M, 1, N, N).unsqueeze(-3).expand(M, 1, N, N, N)
    orientation = torch.zeros((M, 3, 4), device=R.device)
    orientation[:, :3, :3] = R.transpose(1,2)
    grid = F.affine_grid(orientation, V_exp_rotated.size(), align_corners=True)
    V_exp = F.grid_sample(V_exp_rotated,
                          grid,
                          align_corners=True)
    V = V_exp.sum(dim=0, keepdim=True).reshape(N, N, N) / (N * M)
    return V
```

*Code 3.* Python implementation of Algorithm 1.

```python
import torch

@torch.no_grad()
def reconstruction(denoiser, base_image, M = lambda step: 2, t = 0.3, S = 1000, lr = 1,
    t_min = 0.002):
    """
    Input:
        - denoiser: diffusers.UNet2DModel
        - base_image: (N, N)
    Returns:
        - V: (N,N,N)
    """
    # initial volume
    device = base_image.device
    size = base_image.shape[-1]
    min_voxel_val = base_image.min() / size
    V = torch.zeros((1,1, size, size, size), device=device)

    # spherical mask
    mask = create_sphere(size).unsqueeze(0).unsqueeze(0).to(device)

    for step in range(S):

        # 1) forward projection
        rotations = sample_rotations(M(step), device=device, first_ident=w(step) > 0)
        x = forward_project(V, rotations)

        # 2) compute residual
        x_in = x[1:]
        noise = torch.randn_like(x_in)
        xt = x_in + (t**2 - t_min**2).sqrt() * noise
        x0 = denoiser(xt.unsqueeze(1), t).squeeze(1)
        x0 = torch.cat((base_image.unsqueeze(0), x0), dim=0)
        r = x - x0

        # 3) back project residual
        delta = back_project(r, rotations)

        # 4) take gradient step
        V = V - lr * delta

        # 5) enforce support
        V = V * mask
        V[V < min_voxel_val] = min_voxel_val

    return V.squeeze()
```

## B. Experiments

### B.1. Ours

In generating the results presented in this work with Algorithm 1, we deliberately avoided using overly complex schedules or heuristics. Consequently, we kept both the mollification degree $t$ and the learning rate $\alpha$ fixed throughout the reconstruction. The only parameter that depends on the iteration step is the batch size $M$ (i.e., the number of projection images processed per iteration). We observed that starting with a small batch size helps to rapidly improve the initial volume while reducing the risk of becoming trapped in local minima.

In the heterogeneous setting, choosing $M = 2$ also provides a straightforward way to ensure a strong influence of the base image; its relative contribution is higher for small $M$ and diminishes as $M$ increases. In contrast, for the homogeneous case, we increased $M$ linearly from 2 to 50 so that the impact of the base image gradually decreases. This prevents the features of the base image from dominating those introduced by the denoiser. Such a balance is particularly crucial for real data, where

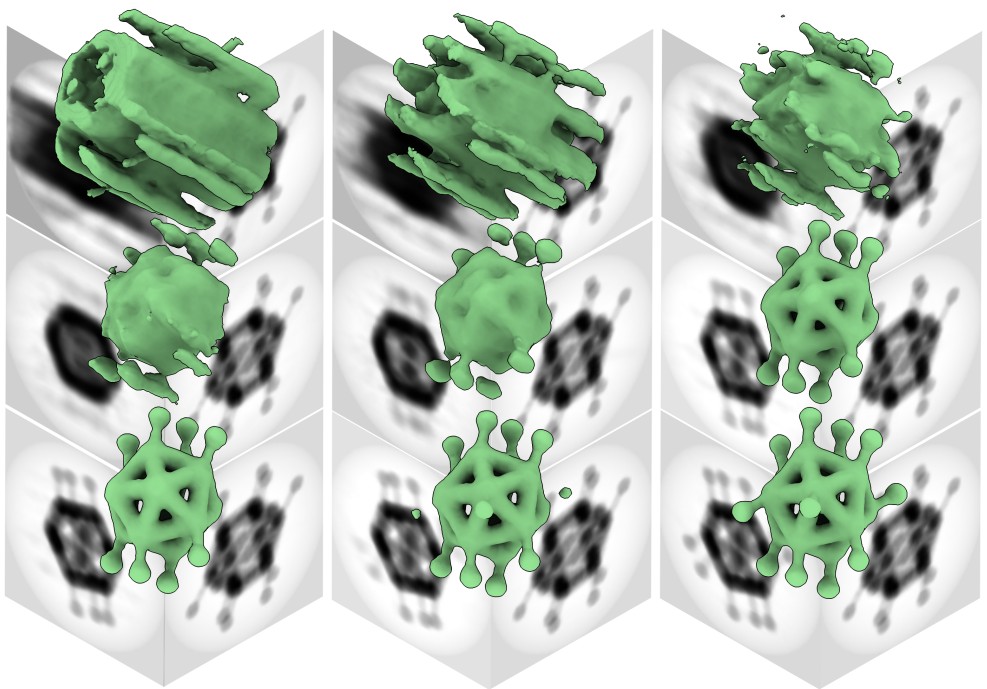

*Figure 8.* From top left to bottom right, our method is illustrated as it incrementally refines the initial volume until the final reconstruction is reached. The volumes shown correspond to iterations $10, 20, 40, \ldots, 2560$. For the base image, the first projection of the dataset was used in this reconstruction.

each projection may contain artifacts. Even in the homogeneous case, the base image remains crucial for stabilizing the reconstruction: without it, the optimization may get stuck in local minima or lead to a rotation of the volume, both of which negatively affect reconstruction quality. For both the reconstruction with Algorithm 1 and the training of the denoiser, we used a large number of iterations to ensure that the denoiser and the volume reconstruction had sufficient opportunities to converge. More details on the parameters used during reconstruction and training are provided in Table 2.

### B.2. CryoDRGN2

We observed that the default CryoDRGN2 training settings are not well-suited for small datasets of about 30–300 images. To address this, we artificially increased the dataset size to roughly 6000 images by repeatedly duplicating the existing images. This simple strategy led to substantially improved results. In addition, we normalized the dataset prior to running CryoDRGN2. We also modified a minor part of the code in the `compute_ctf()` function within `ctf.py`, changing the return statement from `return ctf` to `return torch.ones_like(ctf)`. As a consequence, the CTF operation in CryoDRGN2's forward model effectively became an identity mapping. We continued to supply CTF parameter files `ctf.pkl` for each image as input, but these now serve only as dummy values. We experimented with a range of parameter settings for each dataset and, in some cases, obtained better performance by adjusting the default values of `--lr` and `--feat-sigma`. We also chose a large number of epochs to ensure that CroyDRGN2's training has sufficient time to converge. Nonetheless, in most experiments we primarily retained the default parameters.

*Table 1.* Properties of the test dataset used as input for the reconstruction algorithms.

|  | **NUMBER OF IMAGES** | **IMAGE SIZE** | **DYNAMICS** | **SYMMETRY** |
|---|---|---|---|---|
| BOX-MODEL | 300 | $88 \times 88$ | HETEROGENEOUS | ASYMMETRIC WITH $D_{4h}$ SYMMETRIC CORE |
| CAPSID-MODEL | 200 | $88 \times 88$ | HETEROGENEOUS | ASYMMETRIC WITH ICOSAHEDRAL CORE |
| 80S RIBOSOME | 100 | $136 \times 136$ | HOMOGENEOUS | ASYMMETRIC |
| $\beta$-GALACTOSIDASE | 34 | $64 \times 64$ | HOMOGENEOUS | $D_2$ DIHEDRAL SYMMETRY |

*Table 2.* Parameter we used for the reconstructions per dataset.

|  | BOX-MODEL | CAPSID-MODEL | 80S RIBOSOME | $\beta$-GALACTOSIDASE |
|---|---|---|---|---|
| RECONSTRUCTION |  |  |  |  |
| $M$ (RECONSTRUCTION BATCH SIZE) | 2 | 2 | LINEAR INCREASING FROM 2 TO 50 | LINEAR INCREASING FROM 2 TO 50 |
| $t$ (MOLLIFICATION DEGREE) | 0.2 | 0.5 | 0.3 | 0.3 |
| LEARNING RATE $\alpha$ | 1 | 1 | 1 | 0.5 |
| NUM. ITERATIONS ($S$) | 10K | 10K | 5K | 5K |
| RUNTIME | 2-3 MIN | 2-3 MIN | 2-3 MIN | 2-3 MIN |
| TRAINING |  |  |  |  |
| STEPS | 80K | 75K | 500K | 300K |
| LEARNING RATE | 0.0008 | 0.0007 | 0.0002 | 0.0002 |
| BATCH SIZE | 512 | 512 | 128 | 128 |
| DEVICE | NVIDIA A100 | NVIDIA A100 | NVIDIA A100 | NVIDIA A100 |
| RUNTIME | $\approx$12H | $\approx$11H | $\approx$12H | $\approx$24H |

**Box** `cryodrgn abinit_het box_extended_normed.mrcs --ctf ctf.pkl --uninvert-data --lr 0.0005 -n 300 --checkpoint 100 --zdim 8 --window-r 0.99 --feat-sigma 0.1 -o output_box`

**Capsid** `cryodrgn abinit_het capsid_extended_normed.mrcs --ctf ctf.pkl --uninvert-data --lr 0.0005 -n 300 --checkpoint 100 --zdim 8 --window-r 0.99 --feat-sigma 0.1 -o output_capsid`

**Betagalac** `cryodrgn abinit_homo betagalac_extended_normed.mrcs --ctf ctf.pkl -n 300 --checkpoint 100 --uninvert-data -o output_betagalac`

**Ribosome** `cryodrgn abinit_homo ribosome_crop_extended_normed.mrcs --ctf ctf.pkl -n 300 --checkpoint 100 --feat-sigma 0.25 --uninvert-data --window-r 0.99 -o output_ribosome`

### B.3. SIMPLE

Within the SIMPLE3 GUI, we employed `import_cavgs` and `initial_3Dmodel` to generate the 3D reconstructions.

### B.4. EMAN

For each dataset, we ran the following command to generate 10 candidate reconstructions and then chose the one with the best score: `e2initialmodel.py --input=[dataset-name].mrcs --iter=8 --tries=10 --sym=c1 --automaskexpand=-1 --parallel=thread:5 --verbose=9`

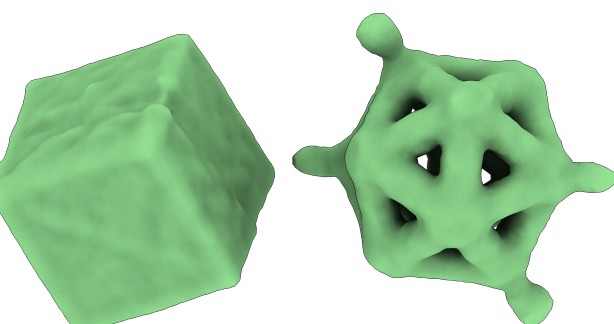

*Figure 9.* Representative failure cases of our method that occur with low frequency. On the **left**, we display the reconstruction generated by the denoiser trained on the box-model dataset, using the 66th image as the base. On the **right**, we show the reconstruction obtained from the denoiser trained on the capsid-model, with 103rd dataset image serving as the base. In both cases, the central core is largely recovered, while the movable box lids and, in the capsid example, most of the arms fail to be reconstructed.

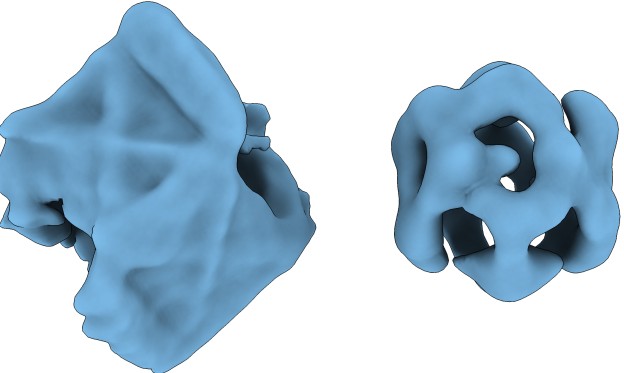

*Figure 10.* Top reconstructions generated by EMAN for the box and capsid datasets.

## B.5. ASPIRE

In the experiments section, the FSC curves were computed using ASPIRE's `.fsc()` function from the `aspire.volume.Volume` class. We generated the reconstruction of the ribosome dataset using the code provided in Code Listing 4.

*Code 4.* Reconstruct Ribosome with ASPIRE.

```python
import numpy as np
import mrcfile
import matplotlib.pyplot as plt
from aspire.source.image import ArrayImageSource
from aspire.abinitio import CLSyncVoting
from aspire.source import OrientedSource
from aspire.reconstruction import MeanEstimator

mrcs_path = "data/ribosome.mrcs"

with mrcfile.open(mrcs_path, permissive=True) as mrc:
    averages = mrc.data.copy()

averages_normed = averages - averages.mean()
averages_normed /= averages_normed.std()

imgs_src = ArrayImageSource(averages_normed[:])
imgs_src_down = imgs_src.downsample(32)
orient_est = CLSyncVoting(imgs_src_down, n_theta=360)
oriented_src = OrientedSource(imgs_src_down, orient_est)
rots_est = oriented_src.rotations
offsets_est = oriented_src.offsets
estimator = MeanEstimator(oriented_src)
estimator.maxiter = 200
estimated_volume = estimator.estimate()
estimated_volume.save('ribosome_recon_aspire.mrc')
```