# OpenReview forum: "Score-based Random Tomography of Dynamic Structures"
_ICML.cc/2026/Conference — Submitted to ICML 2026_

### Official Review · Reviewer_7r1n · 2026-03-12

**Soundness:** 3
**Presentation:** 3
**Significance:** 1
**Originality:** 3
**Overall Recommendation:** 3
**Confidence:** 4

**Summary:**

This paper introduces a novel generative framework for heterogeneous random tomography reconstruction, also referred to as ab initio reconstruction in the context of cryo-EM, which aims to obtain the 3D reconstruction of dynamic, flexible structures from 2D projections under unknown and unobserved orientations. To achieve this, the authors propose a two-stage approach that first trains a score-based diffusion model (a 2D denoiser) directly on 2D class-average images to capture the distribution of object conformations, completely bypassing the need for explicit pose estimation. In the second stage, the method employs score distillation sampling to iteratively optimize a 3D voxel grid for each specific input image. By minimizing the reverse KL divergence, the algorithm ensures that the rendered 2D projections of the optimized volume align with the data distribution implicitly learned by the diffusion model. Ultimately, this mode-seeking strategy successfully decouples 2D representation learning from 3D reconstruction, and the authors demonstrate its capability to recover plausible, continuous 3D structural variations from a sparse set of high-SNR projection images across both synthetic and experimental datasets.

**Compliance With Llm Reviewing Policy:**

Affirmed.

**Final Justification:**

I thank the authors for their detailed rebuttal and follow-up clarifications. After carefully considering their responses, I maintain my recommendation of weak reject.

The authors have clarified their intended scope , pose-free heterogeneous reconstruction of large continuous motions from 2D class averages at intermediate resolution, and I appreciate the transparency. However, my core concerns remain partially unresolved.

The authors argue that their narrow framing (class averages, high SNR, sparse views) is a deliberate and principled choice, and I accept that this is a valid research question. However, the paper does not sufficiently establish why this specific regime constitutes a practically meaningful contribution beyond proof-of-concept. The argument that class averages preserve large-scale conformational variability is reasonable, yet the authors do not demonstrate that the downstream 3D reconstructions are more informative or more reliable than what can be obtained through standard 2D-classification-followed-by-3D-classification pipelines already widely used in the field.

The absence of any comparison to standard practical pipelines makes it difficult to assess the method’s position in the existing landscape. The authors’ justification that such comparisons are outside the scope of the present contribution is understandable, but it weakens the significance claim, particularly for a venue like ICML where empirical validation is expected to be rigorous.

I acknowledge the genuine creativity of the SDS-based formulation and the authors’ honesty in reporting failure cases. I encourage continued development of this direction. However, in its current form, the contribution does not yet meet the bar for acceptance from my perspective.​​​​​​​​​​​​​​​​

**Key Questions For Authors:**

Please see my weakness section.

**Limitations:**

Yes.

**Strengths And Weaknesses:**

### Strengths

1. This is an interesting work, and applying SDS to cryo-EM density map reconstruction is a highly creative approach. The authors focus on a pertinent question in structural biology: how to achieve heterogeneous reconstruction without the need to explicitly estimate camera poses, which I view as one of the most important advantages of this framework.

2. The paper is well-written and easy to follow. The underlying math connecting the diffusion score function to the 3D volume update gradients is solid and well-supported. I also genuinely appreciate the authors' honesty; they clearly present their failure cases, such as the suboptimal performance on the asymmetric 80S ribosome.

### Weaknesses

1. While the theoretical framework is quite novel, applying it exclusively to high-SNR 2D class-averages limits its immediate real-world utility. In practice, cryo-EM heterogeneity is usually resolved directly from tens or hundreds of thousands of noisy, raw particle images, as class averaging often wipes out the subtle continuous dynamics the authors are trying to study. The authors leave scaling to raw particles as future work; thus, as it stands, the submission serves more as a proof-of-concept than a practical tool. As a suggestion, could this framework be combined with state-of-the-art denoising models like Topaz-Denoise or DRACO to reconstruct directly from denoised raw particles?

2. While the paper's central objective pertains to tackling random tomography without pose priors, the current pipeline only operates on a very small set of 2D class-averages (e.g., 34 to 300 images). This sparse dataset size simply cannot densely cover the continuous 3D viewing space. Consequently, the method relies heavily on the structural symmetry of the target to fill in the missing angles. When tested on a fully asymmetric molecule like the 80S ribosome, the method hits a wall, fails to infer the missing views, and ultimately underperforms compared to standard baselines.

3. The experimental setup regarding the baselines is somewhat confusing. For existing ab initio reconstruction baselines like CryoDRGN2, did the authors use the full raw particle stack or just the 2D class-averages? Restricting baselines to a small set of 2D averages might constitute an unfair comparison, as these methods are expressly designed and optimized for large stacks of raw particles.

4. There is no discussion or comparison regarding the runtime or computational cost of the proposed method. Training a 2D diffusion model and subsequently running an iterative, gradient-based reconstruction (via Algorithm 1) independently for every single class-average appears highly computationally expensive. How does the total runtime of this pipeline compare to existing ab initio methods like CryoDRGN2, SIMPLE, or EMAN2? Is it drastically slower? Including a breakdown of training versus reconstruction times is essential for evaluating the method's practical viability.

### Conclusion
Overall, while I do not feel the current set of contributions quite meets the bar for acceptance, I genuinely appreciate the novelty of this framework. I strongly encourage the authors to keep pushing this creative direction and to work toward scaling it up to handle messy, real-world raw particle data in the future.

---

> ### Author Rebuttal · Authors · 2026-03-28
>
> We thank the reviewer for the thoughtful and constructive comments on the practical scope of the method, the realism of the class-average setting, the fairness of the baseline comparisons, and the computational cost; below we address each point in turn and clarify the intended scope and contribution of the paper.
>
> **(W1)** See our response to reviewer ewRr (Q1, W1) for a more detailed discussion of why low-SNR raw-particle reconstruction is outside the scope of the present paper and how we envision this extension in future work.
>
> **(W2)** Thank you for this comment. We agree that reconstruction from a small number of 2D class averages is a challenging setting. However, 34–300 images is precisely the realistic regime for current class-average-based reconstruction tasks, and our goal was to study whether heterogeneous reconstruction is possible in this regime at all. Reconstruction from fewer images is naturally harder than reconstruction from a very large dataset, and this is exactly why we consider the setting nontrivial and worth addressing.
>
> We would also like to emphasize that our framework is not inherently limited to small datasets. In the present paper, we chose to test the method in an extreme and difficult regime in order to demonstrate that it can recover heterogeneous structure even when only a small number of high-SNR inputs is available. E.g., reconstructing 200 distinct 3D conformations from only 200 images in the capsid dataset is a highly challenging task. We are not aware of prior class-average-based methods that achieve reconstructions of comparable quality in such a difficult heterogeneous setting.
>
> Regarding symmetry, we agree that structural symmetry can reduce the number of views required for successful reconstruction in some parts of the object. At the same time, symmetry also creates major ambiguities for traditional methods based on pose search, so it is not exclusively advantageous. More broadly, our results should be interpreted in the context of the problem we target: challenging heterogeneous reconstruction from class averages, rather than standard homogeneous reconstruction from dense coverage of viewing directions.
>
> We also believe it is important to be transparent about limitations and negative examples, especially at a venue like ICML. This is why we included the 80S ribosome experiment. There, our method underperforms relative to strong homogeneous reconstruction baselines, which is consistent with the fact that homogeneous reconstruction is a much easier and much more mature problem with many highly optimized algorithmic solutions. Our focus, instead, is on the harder setting of continuous heterogeneity, where obtaining good initial volumes remains a major challenge and where our approach shows clear promise.
>
> Finally, the current small-data regime should not be interpreted as a fundamental limitation of SDS-based reconstruction itself. Score models are typically trained on very large image collections, often far beyond the scale of cryo-EM datasets, so the framework is not intrinsically tied to having only a few inputs. Extending the approach to larger numbers of high-SNR images, for example by first performing unsupervised denoising of low-SNR data, is a natural next step and aligns with the direction discussed in our response to W1.
>
> **(W3)** Thank you for this comment. For all methods, the input is the same: 2D class averages, not the full raw-particle stack. This is the reconstruction setting studied throughout the paper, and we adapted CryoDRGN2 explicitly to this same class-average regime to make the comparison as direct as possible. We agree that CryoDRGN2 is originally designed for large stacks of noisy raw particles. Precisely because of this mismatch, we adjusted its training in a way intended to help it rather than putting it at a disadvantage: as described in Appendix B.2, we duplicated the small class-average datasets to roughly 6000 images because the default CryoDRGN2 training settings are not well suited for datasets of only about 30–300 images, and this augmentation substantially improved its results.
>
> Finding the correct pose in CryoDRGN2 is key for good reconstructions. In that respect, we do not believe that providing high-SNR images hampers CryoDRGN2; if anything, pose estimation should be fundamentally easier in the high-SNR regime. We agree that CryoDRGN2 tackles a more challenging task in its usual setting, namely reconstruction from low-SNR raw particle images, but we do not think that adapting it to high-SNR class averages makes the comparison artificially unfavorable to CryoDRGN2.
>
> More generally, our comparison should be read narrowly: it is not meant to claim superiority in the standard raw-particle setting, but rather to evaluate how well CryoDRGN2 can be adapted to the specific class-average-based setting considered in this paper. We will clarify this more explicitly in the revision.
>
> **(W4)** See our response to ewRr (Q2, W3).

---

> > ### Author Rebuttal · Reviewer_7r1n · 2026-04-01
> >
> > Thanks for authors' honesty and efforts for rebuttal.
> >
> > **[W1, W2, W3]** The authors' rebuttal is unconvincing for the following reasons:
> >
> > **(1)** The proposed method lacks a practical evaluation setting. CryoDRGN addresses real-world challenges by evaluating continuous heterogeneous reconstruction on raw, low-SNR datasets with pre-computed poses, ensuring a fair baseline comparison. This work fails to provide similar practical benchmarks, specifically lacking comparisons against CryoSPARC's ab-initio reconstruction and CryoDRGN2/DRGN-AI on raw particle stacks. Although the authors argue that their "sparse view + high SNR" scenario is more challenging, they must demonstrate this by comparing it alongside the more standard and practical "dense view + low SNR" setting, which routinely handles massive datasets.
> >
> > **(2)** The motivation for relying on 2D results is fundamentally flawed. 2D classification is designed to filter out bad particles, not to capture subtle 3D conformational changes. Relying directly on 2D results risks losing structural details and blurring conformational states. Have the authors considered the severe entanglement between accurate pose estimation and conformational recovery? CryoDRGN relies on pre-computed poses from well-resolved stacks precisely to break this entanglement, a crucial step that this work seems to overlook.
> >
> > To this end, the specific positioning of this work remains unclear to me. My remaining concerns could be mitigated if the authors could further elaborate on the following: (1) Why is starting from 2D classes a promising and necessary trajectory for high-resolution dynamic reconstruction? The justification should be based on the inherent advantages of this approach. (2) What is the fundamental rationale for avoiding comparisons with standard, practical pipelines or do the authors may accept to compare to them? A clearer explanation of these aspects would significantly aid in understanding the true value and target scenario of this proposed method.

---

> > > ### Author Response · Authors · 2026-04-07
> > >
> > > Thank you for the follow-up. We would like to clarify our intended positioning more explicitly. In the present work, we are not targeting subtle high-resolution conformational variability, but rather large, continuous conformational changes at intermediate resolution. This distinction is important. In cryo-EM, 2D class averages are not used only to discard bad particles; they are also routinely used to characterize structurally meaningful variability, especially large-scale conformational changes, including as part of broader 3D-classification workflows. While very subtle conformational differences are indeed more easily entangled with pose errors and may be blurred by averaging, this is much less true for the large-scale motions that we study here. Our target scenario is precisely the one where large parts of the structure are highly flexible, as in the box and capsid examples.
> > >
> > > This also explains why starting from 2D class averages is, in our view, a promising and necessary trajectory for the present paper. At this stage, class averages provide a practical way to reduce the impact of noise and CTF effects while preserving the dominant conformational variability relevant to our current goal. In other words, they let us isolate and study the heterogeneous random tomography problem itself before solving the additional challenge of reconstruction directly from low-SNR raw particles. We agree that there is a strong entanglement between pose estimation and conformational recovery; however, this is not something our work overlooks, but rather one of the core motivations for our approach. Existing pipelines such as CryoDRGN typically rely on precomputed poses from a well-resolved particle stack in order to break this entanglement. In contrast, the strength of our framework is that it approaches the heterogeneous dataset in a pose independent way: instead of assuming accurate pose estimates in advance, we first learn the distribution of conformations and views represented in the images, and then reconstruct a specific 3D conformation from a given image using the regularized SDS objective. Thus, our goal is not yet to replace standard raw-particle pipelines, but to show that pose-free heterogeneous reconstruction from class averages is possible and promising in the regime of large continuous conformational change.
> > >
> > > This is also the rationale for not comparing directly to standard practical raw-particle pipelines in their usual operating mode. We fully agree that extending the framework to raw-particle pipelines is an important next step, but we do not believe that the absence of such comparisons invalidates the present contribution, because our claim is narrower: we aim to introduce and validate a new approach for unbiased heterogeneous reconstruction of large continuous motions from class averages, not yet to solve the full end-to-end raw-particle problem. We will revise the paper to make this target scenario and positioning much more explicit.

---

### Official Review · Reviewer_uQFg · 2026-03-13

**Soundness:** 2
**Presentation:** 3
**Significance:** 2
**Originality:** 3
**Overall Recommendation:** 4
**Confidence:** 4

**Summary:**

This paper introduces a score-based method of reconstructing dynamic structures from a set of projection images with unknown poses (named “random tomography”). This setup is commonly used in cryo-electron microscopy (cryo-EM), and the merits of the proposed approach is therefore reported through experiments done in synthetic and real cryo-EM datasets. The proposed approach has two components. First is a score-based diffusion model (SBDM) that can capture a probability distribution of images and learn a denoiser to generate images, similar to how usual diffusion models work in other domains. Second is the score distillation sampling (SDS) procedure that utilizes the known differentiable forward model formula of cryo-EM imaging to learn a per-image 3D volume distribution. The volume distribution is guided to the image distribution through the use of reverse KL divergence, which authors highlight as one of the significant changes (in the context of heterogeneity analysis) from the forward KL divergence that the classical ML methods using expectation-maximization routines employ. The proposed method, in its current form, takes only the 2D class averages (and not raw datasets). Therefore, the heterogeneous reconstruction analysis presented here are all based on 2D class averages as inputs. Under this scenario, the results demonstrate the competence of the proposed algorithm with respect to the baselines.

**Compliance With Llm Reviewing Policy:**

Affirmed.

**Final Justification:**

I would like to thank the authors for their clarifications. After extensive deliberation of the text (including appendices) and the rebuttal discussions, I remain reinforced in my borderline recommendation. Although I have significant concerns remaining unsolved about the evaluation (R2), the novelty of the SDS application to cryo-EM and the theoretical maturity of the manuscript makes the potential impact of the work worthy of consideration.

As a result of the discussions, I have increased my significance assessment to 2. The clarifications regarding CTF usage and the compute costs give stronger signal for the usability of the method.

There are multiple reasons why my assessment of the evaluations remain low. The proposed method is situated as a performant solution to an unconventional intermediate setting in the cryo-EM pipeline, i.e. after the class averages are obtained and before the poses are estimated. Unfortunately, there is no empirical evidence provided in support of the existence of this setting, except that it serves as a playground for methods development. One such empirical evidence could have been the demonstration of the superiority of end reconstructions, compared to not using the intermediate setting/method. Another such empirical evidence could have been reduced computational cost in this intermediate reconstruction path. Another evidence could have been a demonstration of how the success of a method in this toy setting indicates higher likelihood of success in the original setting. These remain under-explored, and although I do not consider any specific single one of them as a prerequisite, I consider **at least one** of them as a prerequisite motivation of this setting.

One key aspect of the original cryo-EM setting is the large number of particles. This large number enables redundancy and regularization through ample coverage density on the pose space [SO(3)] and heterogeneity space [e.g., VAE latent space]. When such coverage is not present, e.g. in preferred orientation bias with severely anisotropic pose distribution, the problem starts to fall out of the priors that cryo-EM methods have been designed and developed at. Study of such deviations and method explorations to different priors are open research directions, highlighting the fact that the proposed toy setting is not trivially connected to cryo-EM setting. In the proposed toy setting with 200-300 images, the angular spacing remains sparse; and in the augmented setting for CryoDRGN, the spacing remains the same while being "peaked" at these angles. Furthermore, similar argument could be made by analyzing the sampling sparsity of the heterogeneity angles $(\alpha, \beta)$ of the box dataset and $\gamma$ of the capsid dataset, where conventional methods might be expecting a higher number of samples with hidden benefits of regularization.

Unfortunately, none of these potential empirical studies have been explored and no conclusive theoretical motivation has been made. Therefore, I remain convinced that unless further work is done to connect the setting to the original setting, the significance of the results presented in the manuscript (e.g., Figures 4-7) remain low.

**Key Questions For Authors:**

(Q1) Is it not possible to involve CTF information in your method, so that other methods could also be run in their usual setting?

(Q2) Given the computational demand observed in this toy setting, do you envision an eventual practical application in the standard cryo-EM setting?

**Limitations:**

yes

**Strengths And Weaknesses:**

Strengths

(S1) The paper provides a novel adaptation of diffusion models from first principles to the cryo-EM heterogeneous reconstruction problem.

(S2) The proposed algorithms as well as the experimental procedure for all methods used are clearly documented, supporting reproducibility of the results.

Weaknesses

(W1) This is an unconventional setting, where only the 2D class averages are used for 3D reconstruction. In its current form, especially given the computational costs of the described method (Table 2, 0.5-1 days of A100 for only dozens/hundreds of images), practical application of the method seems highly unlikely.

(W2) The CryoDRGN2 model is run in an unconventional way (Sec. B2): the 2D class averages are augmented through duplication to provide “roughly 6000” images to constitute the input dataset. This is in stark contrast to the usual operational mode, where the complete dataset is provided. Another unconventional choice was to remove CTF correction from the CryoDRGN processing. Representing the volume in the Fourier-space, CryoDRGN might be more upset than the proposed method, which represents the volume in real-space. The authors mention these limitations, but unfortunately they nevertheless significantly undermine the highlighted results in Figures 4-7.

---

> ### Author Rebuttal · Authors · 2026-03-28
>
> We thank the reviewer for the careful reading and constructive comments on the practical scope of the class-average setting, the fairness of the CryoDRGN2 comparison, and the role of CTF information; below we address each point in turn and clarify the intended scope of the paper.
>
> **(W1 and Q2)**
> Thank you for this comment. We agree that our setting is unconventional compared with the more common raw-particle pipeline, but this is also exactly the point of the paper: we study heterogeneous 3D reconstruction from 2D class averages, a regime that is not addressed by existing class-average-based methods, which recover only a single volume, and that differs fundamentally from raw-particle approaches such as CryoDRGN2 or CryoFIRE.
>
> Regarding computational cost, we would like to clarify two points. First, Table 2 already separates denoiser training time from per-volume reconstruction time. The expensive part is the one-time training of the denoiser; after that, the actual reconstruction takes only 2–3 minutes per volume on an A100. Second, in the heterogeneous experiments we reconstruct a large number of distinct conformations: the paper generates 300 box volumes and 200 capsid volumes, one for each class average.   Combining training and inference, this corresponds to roughly 5 minutes per volume for the box dataset and 6 minutes per volume for the capsid dataset.
>
> We also note that, in practice, obtaining cryo-EM data typically requires substantial experimental effort on the scale of weeks to months, so a downstream computational analysis taking on the order of a day is usually not the primary practical bottleneck. For this reason, while runtime is certainly relevant, we do not view the reported cost as making the method impractical for the setting we study. We will clarify this tradeoff more explicitly in the revision.
>
> **(W2)**
> Thank you for this comment. We agree that the adapted CryoDRGN2 setup is unconventional, and we were explicit about these modifications in Appendix B.2. The reason is that CryoDRGN2 is designed for large raw-particle datasets, whereas our setting uses only about 30–300 2D class averages. In this small-data regime, the default CryoDRGN2 schedules are not well suited, and duplicating images to reach roughly 6000 training examples improved its performance substantially. In this sense, the augmentation was introduced in favor of CryoDRGN2, not to weaken it.
>
> Regarding CTF, our inputs are class averages rather than raw particles, and for the CryoDRGN2 comparison we replaced the CTF operation by the identity exactly to better match this setting. As described in the supplement, this was implemented by changing the ``compute_ctf()`` return value to ones, while still supplying dummy CTF parameter files. Since this CTF block is only a linear operator in the forward model and CryoDRGN2 is trained end-to-end with autodiff, we do not believe that this change alone explains the qualitative gap in the heterogeneous results.
>
> More broadly, the fact that CryoDRGN2 represents the volume in Fourier space whereas our method works in real space is an architectural choice, but it does not change the evaluation protocol itself. Both methods are evaluated in exactly the same class-average-based reconstruction setting using FSC after alignment to ground truth.   In the heterogeneous box and capsid experiments, CryoDRGN2 struggles substantially even at low frequencies, whereas our method recovers the conformations much more faithfully.
>
> At the same time, we do not claim that CryoDRGN2 performs poorly across the board. On the 80S ribosome dataset, CryoDRGN2 obtains very good results compared to the state-of-the-art class-average-based reconstruction algorithms, which is consistent with our general point that the comparison was not set up to disadvantage CryoDRGN2.
>
> We will revise the discussion to make even clearer that the CryoDRGN2 adaptation is imperfect but was chosen as a best-effort way to compare against a strong heterogeneous reconstruction baseline in the class-average regime.
>
> **(Q1)**
> Thank you for this comment. In principle, yes: CTF information can be incorporated into our framework. In the present paper, however, we deliberately start from 2D class averages, which are already obtained after standard cryo-EM preprocessing steps including CTF estimation/correction, and we adapt the comparison methods to this class-average setting as well.
>
> Thank you for this important question. For a future end-to-end setup on low-SNR raw data, we see two ways to include cryo-EM-specific constraints such as the CTF. A simple option is to apply a basic correction such as phase flipping before training the denoiser. A more integrated option is to train the denoiser directly on uncorrected raw data, include the CTF explicitly in the forward model, and sample both random rotations and particle-specific CTFs during SGD at inference time. See also our response to reviewer ewRr (Q1, W1).

---

> > ### Author Rebuttal · Reviewer_uQFg · 2026-04-03
> >
> > I would like to thank the authors for their detailed rebuttal. Especially, the clarification that the 2D class averages are after CTF estimation and correction is helpful in justifying the non-CTF setup. The computational cost remarks are also well-received. Unfortunately, I continue to have lingering concerns about the evaluations that prevent me from considering a stronger recommendation.
> >
> > (R1) For not using the CTF variable in CryoDRGN, there seems to be a more [natural way](https://github.com/ml-struct-bio/cryodrgn/blob/a340dcb8804ba1ce76267034a6e7258ea9557329/cryodrgn/commands/abinit_het_old.py#L827) of simply not providing the CTF pickle file as a CLI argument. Is there any specific reason why this was not preferred? If possible, can this be explored?
> >
> > (R2) I still understand the goal of the setting to be the same with the standard heterogeneous ab initio reconstruction (random tomography) case: the experimenter obtains the complete particle stack, and would like to reconstruct heterogeneous maps. I appreciate that the authors see the 2D class averages as an auxiliary “reduction” of the particle stack for this goal, as an intermediate point where their method is situated. However, I believe that the comparison with respect to different approaches should be done in an end-to-end manner. In the synthetic datasets, (1) a full stack should be created, (2) class averages should be obtained/hand-crafted, (3) the result of the proposed method on the averages should be compared with the full stack result of the benchmark methods.
> >
> > This is in alignment with the evaluation of CryoDRGN paper (cf. rebuttal to ewRr), where although CryoDRGN was situated in an intermediate checkpoint, they were able to compete against the standard workflow of the deposited dataset curators. Currently, we cannot disentangle the success of the method from the penalty that the benchmark methods might be suffering due to being only provided with an auxiliary particle stack. This undermines the results provided in Figures 4-7.

---

> > > ### Author Response · Authors · 2026-04-07
> > >
> > > Thank you for the careful follow-up and for acknowledging our clarifications regarding CTF handling and computational cost.
> > >
> > > **(R1)** Not providing the CTF pickle file sets ctf_params=None, which then causes an error later in the code.
> > >
> > > **(R2)** We understand the reviewer’s request for an end-to-end comparison against standard heterogeneous ab initio pipelines on full raw-particle stacks. However, full heterogeneous cryo-EM reconstruction from low-SNR raw data remains a very difficult and still unsolved problem, and we believe such settings should leave room for new ideas that may need several steps to mature before reaching the breadth and utility of established methods. This kind of staged progress is common in the field. For example, the first CryoDRGN paper addressed only a subtask of 3D reconstruction by relying on precomputed particle poses, leaving full ab initio reconstruction to later work.
> > >
> > > Likewise, our present paper does not yet tackle the more difficult low-SNR raw-particle case. Instead, we focus on the already highly challenging problem of high-SNR heterogeneous random tomography from 2D class averages, which we believe is the right setting in which to introduce and validate our approach. In that sense, the controlled comparison in the paper is to provide all methods with the same input representation, namely class averages, and evaluate how well they perform in that regime.
> > >
> > > We agree that this differs from the standard raw-particle heterogeneous reconstruction setting, and we do not claim superiority in that broader end-to-end scenario. At the same time, we do not believe that the benchmark methods are being made artificially weak simply by receiving high-SNR class averages instead of low-SNR raw particles. In particular, for CryoDRGN2, finding the correct pose is key for good reconstructions, and this task should, if anything, become easier in the high-SNR regime. We also adapted its training in a way intended to help it in the small-data setting, by augmenting the inputs to better match its design assumptions.
> > >
> > > Thus, the comparison should be interpreted narrowly: not as an end-to-end contest against standard raw-particle pipelines, but as an evaluation of how existing reconstruction methods adapt to the specific class-average-based heterogeneous reconstruction setting considered here. We agree that an end-to-end comparison on full synthetic raw-particle stacks would be valuable, but we view it as a substantial extension beyond the scope of the current paper rather than a prerequisite for the present contribution.
> > >
> > > Regarding future directions, denoising models have already shown strong unsupervised recovery of clean images from noisy datasets, e.g. Noisier2Noise. In addition, a related anonymized submission currently under review studies the transition from low-SNR to high-SNR images for heterogeneous cryo-EM particle stacks using SDS. Our longer-term goal is to combine such low-to-high-SNR recovery with the present reconstruction framework to move toward full heterogeneous cryo-EM reconstruction from raw particle stacks, potentially even from random micrograph patches without explicit particle picking.
> > >
> > > We will revise the paper to make this positioning even clearer, so that the reader does not interpret Figures 4-7 as claims about superiority in the standard raw-particle setting.

---

### Official Review · Reviewer_ewRr · 2026-03-13

**Soundness:** 2
**Presentation:** 2
**Significance:** 3
**Originality:** 3
**Overall Recommendation:** 3
**Confidence:** 3

**Summary:**

This paper proposes an innovative two-stage framework for 3D reconstruction in cryo-electron microscopy and related imaging fields, specifically targeting cases where projection angles are unknown and the imaging object exhibits dynamic structural changes . The framework cleverly combines score-based generative models with Score Distillation Sampling, breaking the limitations of traditional methods that rely on explicit 3D generative models or heavy angular searches.

**Compliance With Llm Reviewing Policy:**

Affirmed.

**Final Justification:**

The authors have adequately addressed the computational benchmarking (Q2), but the discussion on algorithmic robustness (Q1/W1) and physical reliability (W2/Q3) remains elusive. Explicitly addressing these concerns would be vital to demonstrating the method's true potential and reliability for biophysical research.

**Key Questions For Authors:**

1: Could you add experiments that involve training and reconstruction directly on low-SNR raw data (noisy projections) to verify the algorithm's robustness?

2: Could you provide a more detailed comparison table regarding computational efficiency?

3: Could you discuss how to incorporate physical constraints specific to cryo-EM (such as further integration of CTF correction) to mitigate potential artifacts produced by SDS?

4: Beyond FSC curves, could you include local resolution assessments to better demonstrate the reconstruction quality in dynamic regions?

**Limitations:**

Yes

**Strengths And Weaknesses:**

Strength

1: By decoupling 2D distribution learning and 3D volume reconstruction, the method demonstrates greater flexibility when handling highly heterogeneous datasets.

2: The proposed approach implicitly guides the optimization of the 3D volume toward high-probability projection directions via the score function of the diffusion model, which greatly simplifies the workflow.

3: The use of voxel grids, which aligns with biological imaging conventions, ensures the physical interpretability of the final results.

4: The authors  provided a candid discussion regarding the limitations of their work.

Weakness

1: In a practical cryo-EM pipeline, obtaining high-quality heterogeneous class averages is a massive challenge in its own right. If the method cannot directly handle raw data with extremely low signal-to-noise ratios, its practical utility will be severely limited.

2: In the computer vision community, SDS is known to suffer from over-smoothing or over-saturation issues. In the context of biomolecular reconstruction, this tendency might lead to the loss of subtle structural features or the creation of false geometric configurations. The paper lacks a deep discussion on the risk of hallucinations—whether the model might generate structures that look plausible but are physically non-existent.

3: Using voxel grids for 3D reconstruction means that memory and computational demands grow cubically as resolution increases. The paper does not provide a detailed comparison of the computational overhead between this method and end-to-end approaches like CryoDRGN2.

4: The literature review is relatively limited. While brevity is common in Nature Methods, a more comprehensive survey of related work would provide better academic context for a conference submission.

---

> ### Author Rebuttal · Authors · 2026-03-28
>
> **(Q1, W1)** Thank you for this important comment. We understand the request to already address the full cryo-EM reconstruction problem. However, full heterogeneous cryo-EM reconstruction from low-SNR raw data remains very difficult and unsolved, and we believe such settings should leave room for new ideas that may need several steps to mature before reaching the breadth and utility of established methods. This kind of staged progress is common in the field. E.g., the first CryoDRGN paper addressed only a subtask of 3D reconstruction by relying on precomputed particle poses, leaving full ab initio reconstruction to later work. Likewise, our present paper does not yet tackle the more difficult low-SNR raw-data case. Instead, we focus on the already highly challenging problem of high-SNR heterogeneous random tomography, which we believe is the right setting to introduce and validate our approach. Regarding next steps, denoising models have already shown strong unsupervised recovery of clean images from noisy datasets, e.g. Noisier2Noise. In addition, a related anonymized submission currently under review studies the transition from low-SNR to high-SNR images for heterogeneous cryo-EM particle stacks using SDS. Our longer-term goal is to combine such low-to-high-SNR recovery with the present reconstruction framework to move toward full heterogeneous cryo-EM reconstruction from raw particle stacks, potentially even from random micrograph patches without explicit particle picking.
>
> **(W2)** In our setting, the key question is whether the reconstruction captures the correct coarse conformation and structural variability. Moreover, hallucinated or physically wrong structures are penalized in our evaluation: each reconstruction is evaluated by FSC against its corresponding ground-truth volume after alignment, so false geometric configurations or missing structural parts directly reduce performance. Empirically, our results in the heterogeneous setting are strong under this criterion: the method reconstructs large numbers of distinct conformations for both the box and capsid datasets, and the paper reports FSC-based evaluation across all reconstructed volumes rather than only selected visual examples. We agree that SDS-based refinement toward higher resolution would be very interesting. We will expand the discussion to make clear that, in its current form, our method should be understood as producing heterogeneous ab initio initializations, and that controlling possible SDS-induced over-smoothing at higher resolution is an important future direction.
>
> **(Q2, W3)** We agree that computational efficiency is important. Our appendix reports training time and per-volume reconstruction time for our method: Table 2 (referenced in lines 250, 261, 375). In our measurements, SIMPLE and ASPIRE required 0.5–1h for a single reconstruction on a modern laptop, depending on the number of images, while EMAN required about 1–3h per volume (no GPU support). CryoDRGN2 required about 2h of training on an A100, with roughly 0.4 min/volume for the box dataset and 0.6 min/volume for the capsid dataset. Combining training and inference, our two-step approach requires about 5 min/volume for the box dataset and 6 min/volume for the capsid dataset. We will point more explicitly to Table 2 and expand the discussion of hardware and implementation differences. We also note that acquiring experimental cryo-EM data typically takes weeks to months, so the subsequent computational analysis is usually not the main time bottleneck.
>
> **(Q3)**
> Thank you for this important question. For a future end-to-end setup on low-SNR raw data, we see two ways to include cryo-EM-specific constraints such as the CTF. A simple option is to apply a basic correction such as phase flipping before training the denoiser. A more integrated option is to train the denoiser directly on uncorrected raw data, include the CTF explicitly in the forward model, and sample both random rotations and particle-specific CTFs during SGD at inference time.
>
> **(W4)**
> Our literature review was intentionally focused on the lines of work most directly relevant to the problem setting studied in this paper. Given the page constraints, we prioritized the most closely related methods and the technical context needed to understand our contribution. That said, we appreciate the reviewer’s perspective and will expand the discussion of related work where possible to better position our approach within the broader literature.
>
> **(Q4)**
> Thank you for this suggestion. Our heterogeneous setup differs from a conventional consensus reconstruction setting: each class average corresponds to a distinct conformation, and we reconstruct one volume per image rather than a single global map. We therefore used FSC against the corresponding ground-truth volume as our main metric, since in the synthetic setting it provides a direct and standardized assessment of reconstruction quality for every recovered conformation.

---

> > ### Author Rebuttal · Reviewer_ewRr · 2026-04-04
> >
> > The authors have adequately addressed the computational benchmarking (Q2), but the discussion on algorithmic robustness (Q1/W1) and physical reliability (W2/Q3) remains elusive. Explicitly addressing these concerns would be vital to demonstrating the method's true potential and reliability for biophysical research.

---

> > > ### Author Response · Authors · 2026-04-07
> > >
> > > Thank you for the follow-up. We agree that robustness and physical reliability are central questions. Our present contribution is not a full raw-particle cryo-EM pipeline, but a first demonstration that score-based reconstruction can recover heterogeneous ab initio initializations from CTF-corrected class averages in the high-SNR random tomography regime. In the synthetic experiments, physically wrong or hallucinated structures are directly penalized by FSC against ground truth. We will clarify this intended scope and the current limitations regarding low-SNR raw-particle data and tighter physical modeling more explicitly.

---

### Official Review · Reviewer_PTmW · 2026-03-22

**Soundness:** 3
**Presentation:** 4
**Significance:** 3
**Originality:** 4
**Overall Recommendation:** 5
**Confidence:** 4

**Summary:**

The authors describe a new approach for reconstruction in cryo-EM. Instead of explicitly modeling and optimizing a generative model of 3D volumes from 2D projections, the authors propose 1) learning a score-based diffusion model over 2D projection images (in practice, they use class averages), and 2) an algorithm for 3D reconstruction of a volume V whose projections are consistent with the probability distribution implicitly defined by the learned diffusion model. I enjoyed reading this paper. To my knowledge, this is the first application of score distillation sampling to cryo-EM reconstruction, and the adaptation is non-trivial. The paper is well-written and clear, and the experiments are well-designed.

Some comments and questions:

1) Can the authors comment on the noise level supplied to the denoiser during volume reconstruction? How was 0.3 chosen and how much does it matter?
2) Some additional comparisons would strengthen the manuscript, in particular, comparison against cryoSPARC ab initio reconstruction which also optimizes a voxel array with SGD and cryoDRGN-AI, the more recent version of cryoDRGN2.
3) How important is the soft constraint? Does removing it degrade reconstruction quality, and if so by how much? As a minor comment on semantics, the constraint is not strictly enforced as it enters the loss as a single residual term. Perhaps restraint or alignment penalty would be more accurate.
4) Can the authors comment on the impact of 2D translations in the dataset?
5) The authors need to clarify in the abstract and introduction that the method reconstructs from 2D class averages, not single particle images.

**Compliance With Llm Reviewing Policy:**

Affirmed.

**Final Justification:**

I maintain my score. It seems like some of the other reviewers believe the task is too simple or contrived. While the setting is a bit unrealistic, I think the technical contribution is interesting, and the paper meets the bar for publication as long as the authors make the scope of the work clear (heterogeneous reconstruction of high SNR class averages).

**Key Questions For Authors:**

see above

**Limitations:**

see above

**Strengths And Weaknesses:**

see above

---

> ### Author Rebuttal · Authors · 2026-03-28
>
> We thank the reviewer for the careful reading and constructive questions regarding parameter sensitivity, baseline comparisons, and the role of the soft alignment term; below we address each point and will clarify them more explicitly in the revision.
>
> **1. Can the authors comment on the noise level supplied to the denoiser during volume reconstruction? How was 0.3 chosen and how much does it matter?**
> We keep the reconstruction procedure intentionally simple and therefore fix the mollification/noise level t throughout reconstruction, rather than using a schedule. We chose t=0.3 empirically as a robust default based on preliminary experiments and experience with the method, without performing a dedicated hyperparameter search. In our experience, the method is not very sensitive to the exact choice of t: values in the range of roughly 0.2-0.6 give similar behavior, whereas much smaller values can reduce stability and much larger values can oversmooth the denoiser output. We will clarify this point in the revision.
>
> **2. Some additional comparisons would strengthen the manuscript, in particular, comparison against cryoSPARC ab initio reconstruction which also optimizes a voxel array with SGD and cryoDRGN-AI, the more recent version of cryoDRGN2.**
> Thank you for this helpful suggestion. We agree that additional comparisons are valuable.
> We were not aware of the newest release of cryoDRGN-AI. CryoSPARC ab initio reconstruction is not open-source, so it is not readily adaptable to the class-average-based setting considered in our work. For this reason, we were not able to include it as a direct baseline in a controlled and reproducible way.
>
> **3. How important is the soft constraint? Does removing it degrade reconstruction quality, and if so by how much? As a minor comment on semantics, the constraint is not strictly enforced as it enters the loss as a single residual term. Perhaps restraint or alignment penalty would be more accurate.**
> Thank you for this comment. The soft constraint is important in our method for two related reasons. First, in the heterogeneous setting it biases the reconstruction toward the specific conformation depicted by the given class average, rather than toward an arbitrary high-probability conformation under the learned data distribution. This is exactly the role described in the paper: for each training class average, we reconstruct a separate volume and include a term encouraging the volume’s identity-view projection to align with that observed image.
> Second, the constraint also has a clear stabilizing effect during optimization. As noted in the supplement, the base image remains crucial for stabilization: without it, the optimization can get stuck in local minima or drift to an undesired rotation, both of which reduce reconstruction quality. In our experience, removing this term therefore degrades performance both by weakening conformation-specificity and by making the reconstruction dynamics less stable.
> We also agree with the reviewer’s point about semantics. Since this term is not enforced exactly but instead enters the objective as a residual term, “soft alignment penalty” or “soft restraint” would indeed be more precise than “soft constraint.” We will revise the wording accordingly.
>
> **4. Can the authors comment on the impact of 2D translations in the dataset?**
> Thank you for the question. In general class averages are only approximately centered, so small residual 2D translations are indeed present. To make the denoiser robust to this, we included small random translational shifts as data augmentation during denoiser training. This makes the learned prior less sensitive to minor mis-centering of the class averages and helps the reconstruction remain stable when the input image is not perfectly aligned.
>
> **5. The authors need to clarify in the abstract and introduction that the method reconstructs from 2D class averages, not single particle images.**
> We agree and thank the reviewer for pointing this out. We will clarify this explicitly in the abstract.

---

> > ### Author Rebuttal · Reviewer_PTmW · 2026-04-05
> >
> > Thanks for answering my questions. A quick follow-up on 2: it seems possible to use class averages in cryoSPARC if you provide them as standard single particle images and set CTF parameters to produce the identity function.

---

> > > ### Author Response · Authors · 2026-04-07
> > >
> > > Thank you for the helpful follow-up and for this suggestion regarding cryoSPARC.

---

### Decision · Program_Chairs · 2026-04-30

**Decision:**

Reject

**Comment:**

This paper proposes combining score-based diffusion models with score distillation sampling (SDS) for 3D reconstruction from 2D cryoEM class-averaged images, presenting a novel technical integration in the cryoEM reconstruction pipeline. It is a novel and interesting application of SDS to cryoEM 3D reconstruction.

**Rebuttal summary**
While authors resolved technical questions (e.g., compute efficiency) and agreed to narrow and clarify the scope, they did not adequately address the two significance concerns: (1) the unconventional problem setting (starting from intermediate small-number, high-SNR class averages) lacks justification as a practically meaningful problem; (2) benchmarks adapt ab initio methods outside their intended setting to accommodate the problem, making comparisons questionable.

**Recommendation: Reject** While the technical novelty is acknowledged, the practical relevance of the proposed setting remains unclear even after rebuttal. A future revision should (1) clearly motivate the intermediate class-averaging setting as a practically valuable problem, and (2) provide comprehensive benchmarks against SOTA, including under standard ab initio reconstruction settings.